



# Sensitivity of convectively driven tropical tropopause cirrus to ice habit

Fayçal Lamraoui[1], Martina Krämer[3,4], Armin Afchine[3], Adam B Sokol[5], Sergey Khaykin[6], Apoorva
Pandey[2], Zhiming Kuang[1,2]

[1]Department of Earth and Planetary Sciences, Harvard University, Cambridge, Massachusetts, USA
[2]John A. Paulson School of Engineering and Applied Sciences, Harvard University, Cambridge, Massachusetts, USA
[3]Institute for Energy and Climate Research (IEK-7), Research Center Jülich, Jülich, Germany
[4]Institute for Atmospheric Physics (IPA), Johannes Gutenberg University, Mainz, Germany
[5]Department of Atmospheric Sciences, University of Washington, Seattle, WA, USA
[6]Laboratoire Atmosphères, Observations Spatiales (LATMOS), UVSQ, Sorbonne Université, CNRS, IPSL, Guyancourt,
France

*Correspondence to*: Fayçal Lamraoui (faycal.lamraoui@gmail.com)

**Abstract.** Cirrus clouds that form in the tropical tropopause layer (TTL) can play a key role in vertical transport through the upper troposphere and lower stratosphere, which can significantly impact the radiative energy budget and stratospheric chemistry. The main scientific objectives of this work are to (a) investigate the influence of predefined versus non-predefined categories of ice hydrometeors and ice habit on convectively driven TTL cirrus in simulations of the Asian summer monsoon, and (b) determine the optimal model configuration to adequately simulate TTL cirrus clouds. During the StratoClim field campaign in 2017, aircraft measurements from the southern slopes of the Himalayas captured high ice water content (HIWC) up to 2400 ppmv and ice particle aggregates exceeding 700 μm in size with unusually long residence times. The observed ice particles were mainly of liquid origin, with a small amount formed in situ. The corresponding profile of IWC from the ERA5 reanalysis corroborates the presence of HIWC detrained from deep convective plumes in the TTL but underestimates HIWC by an order of magnitude. We compare the aircraft measurements to a set of numerical experiments at the scales of large-eddy simulations (LES) of the same case study that employ different microphysics schemes. In the TTL, only the scheme that predicts ice habits can reproduce the observed HIWC, ice number concentration, and bimodal ice particle size distribution. The lower range of particle sizes is mostly represented by planar and columnar habits, while the upper range is dominated by aggregates. Large aggregates with sizes between 600 and 800 μm have fall speeds of less than 20 cm s$^{-1}$, which explains the long residence time of the aggregates in the TTL. Planar ice particles of liquid origin contribute substantially to HIWC. The columnar and aggregates habits are in the in-situ range with lower IWC and number concentration. For all habits, the ice number concentration increases with decreasing temperature. For the planar ice habit, relative humidity is inversely correlated with fall speed. This correlation is less evident for the other two ice habits. In the lower range of supersaturation with respect to ice, columnar habit has the highest fall speed. The difference in ice number concentration across habits can be up to four



orders of magnitude, and aggregates occurring in much smaller numbers. We demonstrate and quantify the linear relationship

between the differential sedimentation of pristine ice crystals and the size of the aggregates that form when pristine crystals collide. The slope of this relationship depends on which pristine ice habit sediments faster. Each simulated ice habit is associated with distinct radiative and latent heating rates. This study suggests that a model configuration nested down to LES scales with a microphysical parameterization that predicts ice shape evolution is crucial to provide an accurate representation of the microphysical properties of TTL cirrus, and thus the associated (de)hydration process.


# 1 Introduction

Stratosphere-troposphere exchange plays an important role in the composition of the upper troposphere and lower stratosphere (Dessler and Sherwood, 2004; Stohl et al., 2003). Trace gases such as stratospheric water vapor can directly and indirectly impact climate, radiation (Forster and Shine, 1999; Solomon et al., 2010; Gettelman et al., 2011), chemistry (Dvorstov and

Solomon, 2001) and dynamics (eg. Manney et al 2014). On a global scale, air enters the stratosphere via the tropical tropopause (Brewer, 1949). In the tropics, the transport by the Brewer-Dobson circulation is upward. Therefore, air injected into the lower stratosphere has a longer residence time than into the troposphere and reaches further distances into the stratosphere. The tropical tropopause layer (TTL), which reaches from the tropically averaged level of all-sky zero net radiative heating (14 km, 150 hPa, 355 K) to the upper boundary of 18.5 km (70 hPa, 425 K), is the gateway to the stratosphere (Fu et al., 2007,

Fueglistaler et al., 2009). In addition to large-scale processes, determining how convection may contribute to cross-tropopause transport is essential for understanding Earth's climate (Anderson et al., 2017). To identify the key mechanisms that control the stratospheric water vapor, considerable attention has been given to the role of cross-tropopause convective outflow (e.g., Homeyer et al., 2017; Smith et al., 2017; Ueyama et al., 2018). The Asian summer monsoon (ASM) is the dominant contributor to annual water vapor peaks in the lower stratosphere and the associated Asian summer monsoon anticyclone (AMA) (eg

Fueglistaler et al., 2005; Ueyama et al.,2018; Ploeger et al., 2015). The impact of deep convection on hydration (e.g., Khaykin et al., 2009) and dehydration of the stratosphere (e.g., Sherwood and Dessler, 2001) over the southern slopes of the Himalayas, the Bay of Bengal, and the South China Sea (Fu et al., 2006; Devasthale and Fueglistaler, 2010; Wright et al., 2011; Tissier and Legras, 2016; Legras and Bucci, 2020) is an ongoing debate. The amount of water vapor injected into the stratosphere by overshooting convection is regulated by the cold point temperature (CPT) in a process known as freeze-drying (Avery et al.

2017, Brewer, 1949; Evan et al. 2013; Fueglistaler et al. 2005; Holton and Gettelman, 2001; Randel et al. 2006; Read et al. 2004; Schiller et al., 2009). On the other hand, ice particles injected into the lower stratosphere bypass the cold trap, sublimate, and contribute to stratospheric hydration (Anderson et al.,2017; Herman et al., 2017; Chemel et al., 2009; Khaykin et al., 2009). Cirrus clouds trap heat originating from the Earth's surface, scatter incoming shortwave radiation and affect the intensity of large-scale circulation in the tropics, making them an important regulator (e.g., Randall et al., 1989; Dodion et al., 2008) and

one of the largest uncertainties in the prediction (Boucher et al., 2013) of Earth's climate. Satellite measurements in the tropics





reveal that cirrus cloud cover can be 60–80% at any given time (Baran, 2012), and the net radiative effect of cirrus clouds depends on their macroscopic properties and the size distribution and shape of their ice particles (e.g., Jensen et al., 2009; Lawson et al., 2010; Baran, 2012). A comparison of 19 GCMs revealed that the most significant discrepancies were in the upper troposphere where ice clouds are dominant (Jiang, 2012). Therefore, an accurate understanding of cirrus-related

microphysical processes is vital for predicting climate with confidence (e.g., Schmitt et al., 2016).

Current state-of-the-art microphysical parameterizations are primarily tested and designed for the lower and mid troposphere rather than the cold point tropopause and lower stratosphere, which have very different and complex environmental conditions. The growth of ice particles by vapor deposition in a supersaturated environment is sensitive to the temperature regime, resulting in a distinct ice habit and fall speed (e.g., Phillips et al., 2015). The treatment of ice particles and their shapes in models is very

challenging (Pruppacher and Klett, 2010) and became predominantly independent of measurements (Waliser et al., 2009; Schmitt et al., 2016). Moreover, the microphysical processes of pure ice clouds at these high altitudes are poorly understood due to the difficulty of measuring the key parameters with the required accuracy (Krämer et al., 2016). Most widely used microphysics schemes (e.g., Lin et al., 1983; Thompson et al., 2008) rely on traditional mass–dimensional ($m$–$D$) and terminal velocity–dimensional ($V$-$D$) relationships which have inherent shortcomings and lead to an inaccurate evolution of the density

and shape of ice particles (Sulia et al., 2013; Jensen and Harrington, 2015). Other recent attempts (e.g., Morrison and Milbrandt 2015) included aggregates but without prediction or evolution of ice density. Many recent laboratory (e.g., Järvinen et al., 2018; Harrington and Pokrifka, 2021; Pokrifka et al., 2020) and theoretical (e.g., Przybylo et al., 2019, 2022a, 2022b; Schmitt and Heymsfield, 2010, 2014) studies have examined processes of ice particle growth and provided adequate details on ice crystal complexity, ice habits and aggregates that are overlooked in most sophisticated microphysical parameterizations.

According to ice habit diagrams (e.g., Bailey and Hallett, 2009), the shape of ice can change significantly in response to a temperature change of a few degrees Celsius.

In this study, we explore the ability of commonly used microphysics schemes to reproduce measured peaks of ice water content, ice number concentration, and large ice aggregates in the TTL. We also investigate the impact of different ice habits on fall speed, relative humidity over ice and diabatic heating. Thus, we aim to identify key parameters of these schemes and model

configuration that can lead to an adequate representation of TTL cirrus. For model setup, we stretch our computational resources to its limits and configure a real case study at very high spatial (LES-scale) and temporal resolutions. We compare a set of simulations using different schemes to aircraft measurements from the StratoClim campaign 2017.

The remainder of this paper is organized as follows. Section 2 provides an overview of the StratoClim field campaign and aircraft measurements used in this study. Section 3 presents the numerical experiments, including details on the microphysical

parameterizations involved. Section 4 discusses the results of the model and the comparison with the measurements. Section 5 summarizes the results and presents recommendations for modelers to adequately simulate TTL cirrus.



## 2 Data and methods

### 2.1 Field campaign and observational data sets

The in-situ measurements used in this study are aircraft data from the StratoClim campaign, which took place in the Indian subcontinent from July 27 to August 10, 2017. The M55-Geophysica high-flying research aircraft deployed in Kathmandu, Nepal, was equipped with a comprehensive set of instruments to capture high quality datasets which include but are not limited to, water vapor, number densities, size distributions of ice particles, as well as meteorological parameters in the subtropical upper troposphere/lower stratosphere (UTLS) region (Khaykin et al., 2022).

In this study, we use in-situ measurements of temperature and pressure by TDC (ThermoDynamic Complex) (Shur et al., 2007) as well as measurements of the cloud particle number size distribution in the size range of 3–930 μm diameter at a time resolution of 1 Hz (Krämer at al., 2016; Luebke et al., 2016) by NIXE-CAPS (New Ice eXpEriment: Cloud and Aerosol Particle Spectrometer). The ice water content (IWC) is derived from the particle size distribution with a lower detection limit of ~ 0.05 ppmv (Krämer at al., 2020; Afchine et al., 2018). Gas-phase water was measured with the sampling rate of 1 Hz by means of the Lyman-Alpha hygrometer FLASH (Khaykin et al., 2022).

### 2.2 Case study description

While the aircraft completed eight successful flights during the 2017 Asian monsoon season (Krämer et al., 2020, main article and supplementary material), this study focuses on one specific flight (Flight 8, hereafter F8) on August 10 (Fig.1), which was characterized by a stable anticyclone and intercepted strong overshooting deep convection, high ice water contents and very large aggregates of ice particles that persisted for an unusually long duration. Moreover, this case study is characterized by overshooting of ice crystals into a supersaturated environment above the tropopause (Khaykin et al., 2022), resulting in their rapid growth via uptake of water vapor, sedimentation, and hence irreversible dehydration (Schoeberl et al., 2018).

Figure 1 shows the trajectory of F8 along the southern slopes of the Himalayas and the corresponding time series of water vapor, ice water content (IWC), and altitude. High ice water content (HIWC) peaks up to 2400 ppmv are indicated by yellow and red arrows at 9:40 UTC and 9:50 UTC, respectively. In addition to HIWC, large aggregates of ice particles exceeding 700 μm in size and unusually long residence time are measured during F8. With the use of particle images NIXE-CIPg (Cloud Imaging Probe – grayscale) which is part of NIXE-CAPS, ice particles with diameters between 3 and 937 μm are sized and counted (Krämer et al., 2016; Costa et al., 2017). Evidence of these large aggregates is shown in Figure 2 with 6 snapshots of ice particles from CIPG probe during F8. These crystals have a highly irregular structure due to the inherent randomness in their growth via the process of aggregation (e.g., Schmitt and Heymsfield, 2010).



## 3 Numerical experiments

### 3.1 Model setup

All three simulations of the same real case study associated with F8 were performed using Advanced Research version of the Weather Research and Forecasting Model (WRF-ARW) version 4.2 which is a nonhydrostatic, compressible model that uses
a hybrid terrain-following coordinate system (Skamarock et al. 2008; Powers et al., 2017). The simulations are configured with four one-way nested domains with 9-, 3-, 1- and 0.3 km horizontal grid spacings, encompassing the trajectory of F8 over India (Fig. 1). As our focus is on the TTL, the vertical level configuration has been optimized to achieve 100-m grid spacing for the TTL and high spatial resolutions near the surface and the boundary layer to appropriately simulate fluxes that trigger and sustain deep convection. The 173 vertical levels begin with a 50-m grid spacing near the surface which gradually increases
upward until the level of 6 km to reach a 250-m grid spacing which remains constant upward until the 10-km level. Then, the grid spacing decreases from 250 m to reach 100 m in the layer between the levels of 13 and 19 km. Beyond that, it increases again to gradually reach 900 km at model top of 30 km. A sponge layer is employed in the upper 5 km of all domains to damp out any reflection of waves and prevent their buildup below the model top. We apply spectral nudging for the outer domain D01 above the boundary layer to reduce the distortion of large-scale circulation (e.g., Miguez-Macho et al., 2004; Lamraoui
et al., 2018). Due to the limitation of storage and computational resources, each of the simulations is a 12-h integration from 0000 UTC to 1200 UTC 10 August 2017 with a 5-min output frequency. For the analysis, we consider only the last three hours which correspond to the time when the aircraft captured peaks of IWC and large ice aggregates. The simulations were driven using the fifth-generation European Centre for Medium-Range Weather Forecasts reanalysis (ERA5) fields (Hersbach et al. 2020).

Ideally, the WRF-ARW simulations could be compared not only to the Stratoclim aircraft measurements but also to concurrent satellite observations of cloud properties, such as those offered by the CloudSat (Stephens et al., 2002) and CALIPSO (Winker et al., 2009) satellites. While CloudSat and CALIPSO did pass over the study region on the same day as Stratoclim F8 (Figure 3), they did so nine hours after F8. Because this time gap exceeds the convective and microphysical timescales relevant to our analysis, a comparison to CloudSat or CALIPSO retrievals would not be appropriate. While geostationary satellites offer
measurements that are concurrent with F8 in both space and time, they cannot provide vertically resolved retrievals of cloud microphysical properties and would therefore be of little utility here. We thus focus solely on comparing the simulations to F8 and on the sensitivity of the simulations to the choice of microphysics scheme.

Except for the microphysics scheme option, the simulations are carried out using the same model configuration, which includes the Rapid Radiative Transfer Model for General circulation models (RRTMG; Iacono et al., 2008) for longwave and shortwave
radiation, the Eta Similarity Scheme (Janjic, 2002) for the surface layer, the Unified Noah Land Surface Model (Tewari et al., 2004) for land surface physics and the Shin-Hong Scale–aware Scheme (Shin and Hong, 2015) for the planetary boundary layer (PBL). The New Tiedtke Scheme (Zhang and Wang, 2017) for the convection parameterization is enabled only with the outer domain.


### 3.2 Model microphysics

We investigate the sensitivity of TTL cirrus clouds to microphysical parameterizations with predefined and non-predefined frozen hydrometeor categories, as well as to ice morphology. For this reason, we use the following three bulk microphysics schemes (BMS): the aerosol-aware Thompson scheme (Thompson and Eidhammer, 2014), the predicted particle properties (P3) scheme (Morrison et al., 2015; Milbrandt and Morrison, 2016) and the Jensen Ice-Spheroids Habit Model with Aspect-Ratio Evolution (Ishmael) Scheme (Jensen et al., 2017). Size distributions of hydrometeors in BMS are assumed to be a

generalized gamma function. The Thompson and P3 schemes use simple tunable power laws to relate mass and fall speed to the size of each hydrometeor. Schemes with a free-category approach of frozen hydrometeors such as P3 and Jensen are advantageous over classical multicategory schemes (e.g., Chen and Tsai, 2016; Thompson and Eidhammer, 2014; Morrison et al., 2009; Milbrandt and Yau, 2005) that require arbitrary and artificial conversion between categories, which is inherently problematic (e.g., Milbrandt et al., 2021). The Thompson scheme is quasi-two-moment scheme that predicts the mass mixing

ratios of cloud water, rain, cloud ice, snow, and graupel, as well as the number concentrations of cloud ice and rain. Artificial transitions between predefined ice categories and the corresponding ad-hoc conversion parameters can systematically bias the model output (Morrison and Grabowski, 2008; Lin and Colle, 2009). In addition to mass and number concentration, P3 predicts several other bulk physical ice properties such as rime mass, rime density and mean dimension. In P3, the changes of riming state and rime density are transitional but the growth of ice particle by vapor deposition remains constrained by mass–

dimensional (m–D) and fall speed–dimensional (vt–D) relationships that are chosen a priori as in traditional schemes. With the use of m–D and vt–D relationships, it is assumed that all modeled ice particles evolve and sediment along the same trajectory in mass–size and fall speed-size spaces. These relationships do not capture the proper transitions in ice particle mass and fall speed that occur during riming (Jensen and Harrington, 2015). The Ishmael scheme predicts the evolution of ice properties, including aspect ratio, mass mixing ratio, number concentration, size, and ice density. Ice particles are modeled as

spheroids and the nucleation temperature controls the number of nucleated species. The evolution of ice shape during vapor growth is verified against wind tunnel data (Jensen and Harrington 2015). Unlike the other two schemes, this scheme simulates measured habit-dependent riming (e.g., Fukuta and Takahashi, 1999) and its effect on fall speed. The evolution of ice shape from unrimed to lightly rimed to densely rimed is modeled without converting ice mass between predefined ice categories. The shape of the ice particles is controlled by the riming rate, resulting in a reduced vapor growth rate and an increased riming

rate compared with traditional BMSs (Jensen and Harrington, 2015). Light riming can expand the minor axis of ice crystals or fill in gaps in their shapes, leading to increased density and fall speed without affecting maximum dimension. The simulated ice habit is determined based on the growth rate which mainly depends on the capacitance and ventilation coefficient (e.g., Liu et., 2003). This realistic process conflicts with the widely used m–D and vt–D relationships.



## 4 Results and discussion

### 4.1 Convective activity and HIWC at the TTL

Observations have indicated strong convective activity and aircraft-captured peaks of HIWC detrained at the TTL level (Khaykin et al., 2022). We therefore opt for two sanity checks. First, we want to ensure that the model can reproduce the observed convective activity. For this reason, the simulated outgoing longwave radiation (OLR) corresponding to the time of the first HIWC peak measured at 0940 UTC is shown in Figure 5. Multiple cells of very low OLR are simulated reflecting strong convective activity. Second, the aircraft was at an approximately constant level of 17 km when the HIWC values were detected. The corresponding IWC profile is used to provide evidence of HIWC plumes in the TTL. Therefore, we use IWC from ERA5 reanalysis within the inner domain encompassing the trajectory of F8. The 99$^{th}$ percentile of IWC and domain-mean temperature profiles are shown in Figure 6. Although the ERA5 HIWC is about an order of magnitude lower than the measurements (Fig.6), it identifies the presence of a HIWC plume at ~125 hPa in the TTL. The profile of domain-mean temperature shown in Figure 6 indicates a CPT of 192 K at the pressure level of 100 hPa which is about 5 K warmer than the measured.

### 4.2 Sensitivity of simulated microphysical properties to model configuration

In this section, we aim to test the ability of the model to reproduce the microphysical properties observed during F8, including HIWC peaks, ice number concentration and large ice particle aggregates. To examine the sensitivity of simulated HIWC to changes in model grid spacing and microphysical parameterizations, we performed a set of simulations configured with four nested domains down to 0.3-km grid spacing and three microphysics schemes: Thompson, P3 and Ishmael. All simulations share the same high vertical spatial resolution of 100 m in the TTL (Fig.4). HIWC in the upper troposphere and TTL, shown in Figures 7, 8 and 9, indicates sensitivity to grid spacing that varies from one scheme to another. In response to overshooting deep convection, the simulated HIWC profiles shown in these figures reveal a gradual increase over 9 hours.

Despite the high spatial resolution in the TTL, the Thompson scheme (Fig.7) simulates a detrained HIWC in the TTL at ~ 9:00 UTC but with significant underestimation of more than an order of magnitude relative to the HIWC captured by aircraft at 9:40 UTC (Fig.1) and even that of ERA5 (Fig.6).

For the P3 scheme (Fig.8), the IWC profiles show no significant change between the 9- and 3-km grid spacings. On the other hand, with 1- and 0.3-km grid spacings, the difference becomes significant, especially the accelerated increase in the TTL which is higher in the inner domain. This scheme simulates very high IWC peaks between 8:00 and 10:00 UTC that exceed 3000 ppmv, which is overestimated compared to the measured IWC peak of 2400 ppmv at 9:40 UTC.

The Ishmael scheme (Fig.9) shows improvement of simulated IWC only with 1-km grid spacing like P3 but with less accelerated increase in the TTL. What is unique about the Ishmael scheme in this simulation among the other two schemes is that it can approximately reproduce the IWC peak that is observed during F8 at the same time ~09:40 UTC. No overestimation of HIWC is simulated before ~09:40 UTC or after ~10:00 UTC.





The temporal evolution of the storm is well captured by the model and the simulation reproduces strong deep convection cells (Fig.5) in the domain D4 comparable to those observed during the field campaign at the same time. Therefore, for the comparison with F8, we use the simulation data of the entire inner domain D4 which encompasses most of the F8 trajectory.

Following Krämer et al. (2016, 2020), we examine the measured and simulated microphysical properties of cirrus clouds in the TTL using the ice water content–temperature (IWC–T) parameter space (Figs 10, 11) and ice particle sizes larger than 3 µm. The IWC-T figures enable distinguishing the in-situ origin from the liquid origin cirrus clouds. These two types of cirrus clouds are roughly separated by the median IWC (green line, Schiller et al., 2008) in Figures 10 and 11. Above the green line, the ice particles are of liquid origin and strongly ventilated upwards by deep convection. Below this line, ice particles are

associated with weak updrafts and represent mostly in-situ cirrus clouds that are likely heterogeneously, or, less likely, homogeneously (Krämer et al., 2016; Cziczo et al., 2013; Jensen et al., 2013). Comparison between the simulation and the observation in Figures 10 and 11 reveals that most of measured ice particles are of liquid origin with only a small amount in the in-situ range. In-situ simulated cirrus clouds in all schemes (Figs. 10,11) are associated with low values of IWC and number concentration, which makes them less influential in regulating HIWC. These two figures support the results shown in Figures

7, 8 and 9 that Ishmael, P3 and Thompson schemes yield reasonable, overestimated, and underestimated values of HIWC, respectively.

For all schemes, higher ice number concentrations are associated with ice particles of liquid origin and strong updrafts. Higher values of IWC are mostly in the colder temperature range. In the temperature range between 187 and 192 K shown in Figures 10 and 11, the ice number concentration is correlated with IWC, as discussed by Kramer et al. (2020, their Figure 6). The

simulated ice number concentration has a reasonable upper limit of about 10 cm$^{-1}$ which is comparable to observations (Fig. 12). The measurements indicate colder temperatures reaching 187 K, which is not the case for the simulations, in particular with the Ishmael and P3 schemes.

The results with the Ishmael scheme shown in Figures 11 reveal that the ice particles of liquid origin and which contribute substantially to HIWC are of planar shape. Columnar and aggregate habits are mostly in the in-situ range with very low IWC

and number concentration values.

Also, planar ice particles have the highest number concentration among the three habits, especially at colder temperatures (Fig. 12). For all habits involved in this study, the number concentration increases with decreasing temperature (Figs. 11and 12), which corroborates previous studies (e.g., Gryspeerdt et al., 2018; Sourdeval et al., 2018).

In addition, the distribution of ice number concentration shown in Figure 13 exhibits a large spread across ice habits that can

reach four orders of magnitude. The planar habit takes the lead in terms of high ice number concentration with a 99$^{th}$ percentile of 8 cm$^{-3}$, followed by the columnar and then aggregates with the lowest number concentration. As expected, aggregates are much less frequent than other habits (e.g., Lawson et al., 2006).

In the comparison illustrated in Figure 14, we examine the relative humidity with respect to ice (RHi) which is a requisite environmental condition necessary for the nucleation and growth of ice crystals (Pruppacher and Klett, 2010; Heymsfield and

Miloshevich, 2013). In the TTL, the supersaturation with respect to ice can be reached through different processes such as



moisture lifting by strong updraft (Evans and Marchand, 2013) or temperature fluctuations due to gravity waves (Haag et al., 2003). In the inner domain, RHi saturation is mainly caused by moisture uplifting induced by deep convection. This comparison demonstrates that the CPT from the P3 and Thompson schemes are at potential temperature levels of 380 K and above. On the other hand, the Ishmael scheme produces a CPT at the level of 375 K which is closer to the observed level of

~372 K.

Figure 14 shows that around the CPT, the Ishmael scheme produces larger differences between the RHi inside and outside the clouds than the other two schemes. Furthermore, Ishmael indicates very high values of RHi around CPT, which favors an enhanced freeze-drying process and further in-situ and homogeneous nucleation. Unfortunately, at and around the CPT level, the observation does not provide enough data outside clouds for an adequate comparison with the simulations.

**4.3 Ice particles size distribution**

In terms of ice particle sizes, Figure 15 shows that all simulated effective radius distributions are underestimated compared to the observations, in which the 99[th] percentile is 131 μm. The 99[th] percentiles of effective radius simulated using the Ishmael, P3 and Thompson schemes are 70 μm, 57 μm and 25 μm, respectively. The Ishmael scheme produces a bimodal distribution with two peaks at about 5 μm and 25 μm which have the highest frequencies.  For the smaller peak, this can be explained by

freshly nucleated ice particles due to high RHi (Fig. 14) that are of in-situ origin. On the other hand, for the larger peak it can be due to the rapid growth of existing particles that are of liquid origin before sedimentation (Fig. 11). The distributions of the Thompson and P3 schemes are shifted toward smaller sizes.

The distribution of P3 dominates the range of small sizes. In combination with a higher CPT level and lower RHi saturation (Fig.14), ice particles are more likely to grow slowly, resulting in lower fall speeds, longer residence time and higher IWC.

This explains the overestimation of the simulated IWC using the P3 scheme shown in Figures 8 and 10.

In this section, we focus on the simulation with the Ishmael scheme to examine the sensitivity of TTL cirrus clouds to different ice habits. We use the maximum dimension $D_{max}$ (Jensen et al., 2017), which is produced only by Ishmael scheme among all the schemes involved in this study. The observed ice particles shown in Figure 2 illustrate very large sizes captured during F8. For a fair comparison with the images captured during the field campaign, Figure 16 shows the distribution of $D_{max}$ from

measurements and the Ishmael simulation. It demonstrates the bimodality (e.g., Ivanova et al., 2001; Mitchell et al., 2008; Zhao et al., 2011) of the observed and simulated cirrus particle size distributions (PSD), with an approximate 300-μm shift of the simulated size range towards larger values. The small particle mode is in the sub-100-μm and the sub-400-μm size ranges for the observations and simulation, respectively. The bimodality of the PSD over a very wide range of particle sizes indicates a difference in the physics controlling its evolution (Field and Heymsfield, 2003; Sourdeval et al., 2018). Bimodal cirrus PSDs

can be explained by aggregation processes and depositional growth. Depending on RHi, bimodality evolves naturally due to condensational growth and sedimentation (e.g., Mitchell et al., 1996). It can be difficult to distinguish aggregation from complex diffusional growth patterns (e.g., Lawson et al., 2006), using only in-situ observations of microphysical properties. The lower range of particle size mainly represents nucleation and diffusional growth, while the higher range is dominated by



aggregation (Field, 2000). The simulated PSD shown in Figure 16 provides additional details that distinguish the different ice
habits. The first simulated mode of the PSD is dominated by the planar habit followed by the columnar one with a narrower
range for sizes <300 $\mu$m. The second mode, however, is only associated with the aggregates that dominate the large end of the
PSD (Kajikawa and Heymsfield, 1989; Heymsfield et al., 2002). The large end of the observed PSD > 400 μm shares the same
range of simulated aggregates. Observed ice particles with sizes larger than 700 μm only account for 6.4 %, which corroborates
the low ice number concentration of the simulated aggregates shown in Figures 11, 12 and 13. The maximum size of the ice
particles captured by F8 is approximately neighboring the simulated median and the most frequent size range of the simulated
aggregates. More specifically, in the anvil region, the median value of aggregate sizes is 828 μm, which corresponds
approximately to the maximum value observed during F8. Moreover, the simulated PSDs for all ice habits shown in Figure 16
demonstrate a difference between ice particle sizes in the convective and anvil regions. To distinguish between the three life
cycle stages of deep convection, three ranges of ice water path are used: deep convection (IWP>1000 g m$^{-2}$), thick anvil
(1000>IWP>10 g m$^{-2}$), and thin cirrus (10>IWP>0.1 g m$^{-2}$) (Sokol and Hartmann, 2020). In reference to PSDs in the convective
region, the larger end of the PSD in the anvil is smaller for the columnar and aggregate habits. For the planar habit, Figure 16
shows a broadening of the distribution in the anvil. Despite the overestimation of the simulated particle sizes, the Ishmael
scheme is the only scheme used in this study that can reproduce the large particle sizes captured by F8 and identify their shape.
While the range of particle sizes for each habit is similar in anvil and deep convective clouds, the largest particles are more
frequently found within deep convection.

During F8, it was reported that very large aggregates of ice particles were observed for an unusually extended long time without
falling. For this reason, we want to check whether this tendency is also reproduced by the simulation. Figure 17 shows the
time series of the change in the median value of the maximum dimension for the three ice habits, in reference to the initial
value of August 10, 2017, at 9:00 UTC. The temporal evolution of the aggregates size indicates no decrease in the median
value during the whole three hours (Fig.17), which backs up the observed persistence of the large aggregates. The size of the
columnar habit is similarly constant, fluctuating by less than 2%. On the other hand, the size of planar ice particles experiences
an approximately constant decrease of 4% per hour. From the PSDs of each ice habit shown in Figure 16, the planar shape
occupies a wide range that envelops that of the columnar habit. The large planar ice particles either fall off if they become
sufficiently heavy or contribute to the formation of aggregates. The lower end of the PSD for the planar habit represents mostly
small, freshly nucleated ice particles. Moreover, due to the difference in RHi, the PSD indicates more in-situ nucleation of
planar ice particles in the anvil than in the convective region (Fig.16). Therefore, the steady decrease in particle size for the
planar habit shown in Figure 17 is due to the loss of large ice particles in favor of sedimentation or aggregate formation and
the increasing nucleation of small ice particles. For the columnar habit, ice particles contribute to aggregate formation and
experience growth or sublimation depending on the RHIi. The low rate of in-situ nucleation prevents the median value of the
maximum dimension of the columnar habit from experiencing a strong decrease undergone by the planar habit (Fig. 17). Due
to the coexistence of columnar and planar habits, ice aggregates form continuously. The aggregate habit has a hollow structure
with a very low density of ice particles more than an order of magnitude smaller than the density of the planar and columnar





habits. For this reason, ice aggregates can remain longer in the TTL even with large sizes until reaching the critical sedimentation size.

## 4.4 Fall speed and growth of ice particles across ice habits

The fall speed of ice particles is one of the factors that determines the lifetimes of cirrus clouds (e.g., Jiang et al., 2019) and affects climate sensitivity (e.g., Sanderson et al., 2008). The corresponding RHi which can vary significantly (Fig. 14) in the TTL regulates the rates of nucleation, sublimation, and growth of ice particles. Due to the difference in shapes and densities, ice particles can evolve differently in the outflow of deep convection, anvil, and in-situ thin cirrus (e.g., Lawson et al., 2019). To further explore the behaviour of each ice habit in these three regions in the TTL, Figure 18 shows scatter plots of RHi as a function of fall speed, color coded by $D_{max}$. The StratoClim field campaign and more specifically the F8 trajectory aimed to sample deep convective cases. The results of the inner domain D4 (Fig. 3) using the Ishmael scheme shown in Figure 18 are consistent with the observation that indicates the dominance of deep convective and thick anvil clouds, with a very low occurrence of thin cirrus (Fig.18). The bottom row of Figure 18 shows the small amount of thin cirrus consisting exclusively of planar ice habit with small sizes mostly less than 100 μm and low fall speeds below 20 cm s$^{-1}$. For higher values of RHi ~130% and above, these planar ice particles are largely formed in situ, and are mostly 50 μm or smaller with fall speeds of ~10 cm s$^{-1}$. These small ice particles explain the extended life cycle of thin cirrus clouds.

For deep convection and thick anvil regions, Fig.18 (upper and middle rows) shows the coexistence of the three ice habits but with different characteristics. The dependence of fall speed on particle sizes is not similar from one ice habit to another, which is mainly due to the difference in particle shape and density. In terms of fall speed, the deep convection region within the TTL occupies the largest range starting from a few centimetres per second for planar ice to over 150 cm s$^{-1}$ for columnar ice. For planar ice, when RHi<130%, Figure 18 shows the smallest ice particles (<30 μm) with very slow fall speeds (<5m s$^{-1}$). These particles are mainly of liquid origin and freshly nucleated. On the other hand, planar ice particles associated with higher supersaturation (RHi>150%) are more likely to grow and fall in the deep convection region (upper left). In the anvil (middle left), high supersaturation (RHi>150%), very low fall speed and small particle size indicate prevalent in-situ nucleation. Also, for the planar habit, RHi is inversely correlated with fall speed. This correlation is less evident for the other two ice habits. For the columnar ice habit, Figure 18 (middle column) shows a shift in fall speeds to higher values, larger particle sizes and lower supersaturation with respect to ice. These characteristics indicate slower growth and high sedimentation rates when compared with planar habit. Larger particles are more likely to sediment with fall velocities > ~1 m/s (Heymsfield and Wright, 2014) from the outflow depending on ice density, as soon as they are no longer supported by an updraft. As ice particles sediment. they may contribute to aggregate formation, grow or sublimate depending on the RHi. High fall speeds and RHi > ~130% such as experienced with columnar habit (Fig.18), may correspond to regrowth downwind of aged ice particles via vapor diffusion (e.g., Gallagher et al., 2012). In contrast, the aggregates demonstrate different behavior than pristine columnar and planar crystals. Very large particle sizes between 600 and 800 μm are associated with very low fall speeds below 20 cm s$^{-1}$. This



explains the longer residence time of aggregates observed during F8. For these ice particles to fall out with speed of about 30 cm s$^{-1}$ or faster, they need to reach a size of ~900 μm or larger.

As ice crystals sediment, they may aggregate (Lawson et al., 2019) by differential sedimentation throughout the cloud layer (e.g., Schmitt and Heymsfield, 2010). Large ice crystals with higher fall speeds can collect smaller ice crystals as they fall through clouds and form aggregates. The ratio of the sizes of aggregating crystals and their terminal velocities are important

in aggregation (e.g., Kajikawa and Heymsfield, 1989). To explore this relationship and quantify the correlation between maximum dimensions and corresponding fall speeds, Figure 19 shows density plots that relate these two parameters. The figure reveals the linear relationship across all ice habits and demonstrates comparable shallow slopes of about 3 μm/cm s$^{-1}$ for columnar and planar habits, but with a larger spread for the latter. For columnar ice particles, Figure 19 (middle column) shows a sharp correlation. A small change in size can lead to a large change in fall speed, which is supported in Figure 18 (upper row,

middle column) by higher fall speeds. The large spread associated with the planar habit is due to the larger aspect ratio range. The minor axis can evolve broadly and regulates ice crystal density and fall speed without affecting the maximum dimension. On the other hand, the aggregates indicate a sharp correlation and a steep slope of about 27 μm cm s$^{-1}$. This reveals a lower sensitivity of fall speed to change in maximum dimension which is due to very low ice densities for this ice habit.

To explore the relationship between differential sedimentation and maximum dimension ($D_{max}$) of aggregates, Figure 20

reveals an evident linear relationship between the $D_{max}$ of aggregates and the difference in fall speeds between two different ice habits (columnar-planar) throughout the cloudy TTL. This relationship is consistent in the deep convection (Fig.20, right) and thick anvil (Fig.20, left) regions. Aggregates can also occur involving similar ice habits, but these density plots (Fig.20) reveal a predominance of aggregation formation through different habits in the controlling the maximum dimension of aggregates. Also, Figure 20 provides additional details on the differential sedimentation associated with the case of the current

study. In the formation of aggregates, which ice habit falls faster is important in determining the slope and the spread of the linear relationship. When the planar habit sediments faster than the columnar, the slope of the linear relationship is steep of about 28 μm/cm s$^{-1}$ and the linear relationship is sharp. On the other hand, when the columnar habit sediments faster than the planar, the slope of the linear relationship is even steeper at ~87 μm/cm s$^{-1}$, and the maximum dimension of the corresponding aggregate is more sensitive to the difference in fall speeds. The spread of the linear relationship in this case is larger.

**4.5 Sensitivity of diabatic heating in TTL cirrus to different ice habits**

Although this study focuses on the sensitivity of HIWC in the TTL to model configuration and different ice habits, we extend our analysis in this section to explore the sensitivity of diabatic heating to ice habits. We examine the sensitivity of radiative and latent heating rates to different ice habits, spanning hours 9–12 UTC. The heating rates in Figure 21 show distinct distributions between different cloud morphologies (thick anvil and thin cirrus), as well as between ice habits. Figure 21 (left

column) shows the planar habit with an approximately balanced distribution with an average radiative heating rate of 1.5 K day$^{-1}$. Moreover, moving away from deep convection, the radiative heating rate associated with planar habit gradually shifts towards more positive radiative heating rate in thick anvils with an average of ~ 2.5 K day$^{-1}$ and even more in thin cirrus with





average of ~ 3.5 K day⁻¹. The radiative heating rate associated with thin cirrus clouds is predominantly positive. If only planar habit is considered, this tendency is consistent with the findings of Hartmann et al. (2018) that rising and thinning anvils lead

to a net radiative warming effect. Adding more realistic and complex ice habits in the microphysical parameterizations can alter the radiative heating of clouds. For columns and aggregates, the distributions of radiative heating rate are slightly different, with averages of 4 and 4.6 K day⁻¹ in deep convection and 4.6 and 4.3 K day⁻¹ in thick anvil, respectively.

Comparison between the three ice habits indicates that the columnar habit takes the lead in net radiative warming, followed by the aggregates, then the planar. The contribution of the columnar habit versus the planar habit to the net radiative rate is more

than double in the deep convection region and almost double in the thick anvil. Therefore, neglecting the shapes of ice particles can lead to an underestimation of the net cloud radiative rate by more than double.

Latent heating rates (Figure 21, right column) also indicate sensitivity to ice habit and cloud morphology. All distributions are skewed to positive values, except for the planar habit which indicates an approximate balanced distribution in the deep convection region. Across different stages of cloud morphology, the positive skewness of the latent heating distributions

reflects a predominance of ice particle growth and nucleation. In the deep convection region and within the TTL, columnar habit growth and sublimation is dominant over the other two habits, with average latent heating rate of 2.3 K day⁻¹. The average values for the planar and aggregates habits are 1.2 and 2.1 K day⁻¹, respectively. The transition from deep convection to thick anvil indicates a significant narrowing of the latent heating rate distribution associated with columnar habit and decreasing its average value to 1.52 K day⁻¹. The decrease in the rate of sublimation and the growth of columnar habit is consistent with the

higher sedimentation shown in Figure 18 (middle column). Also, Figure 21 (right column, middle row) shows dominant positive heating rates associated with aggregates which reflects continuous aggregates growth in thick anvil. The average latent heating rate associated with aggregates in the anvil is 1.54 K day⁻¹. For thin cirrus clouds with an average latent heat of 0.1 K day⁻¹, Figure 21 mainly shows lower positive latent heating rates with negligible negative values. Insignificant sublimation and low growth rate will allow ice particles to persist without reaching the sedimentation threshold, resulting in a prolonged

residence time in the TTL. This supports the continued nucleation of small ice particles and the slower growth rate demonstrated in Figures 17 and 18. Comparison of latent warming associated with the three ice habits indicates that aggregates and columnar habits contribute more than twice as much as planar habit. For diabatic heating in thick anvils, columnar habit dominates the net radiative warming and aggregates dominate net latent warming. The results shown in Figure 21 quantitatively confirm the major impact that different ice particle habits can bring to diabatic heating and consequently to TTL cirrus life

cycle.

## 5 Summary and conclusions

Many previous studies have addressed the impact of ice habit in mixed-phase clouds (e.g., Avramov and Harrington, 2010; Hashino et al., 2020; Harrington et al., 2013; Sulia et al., 2021), but fewer have investigated its impact on TTL cirrus clouds, which occur in significantly different environments. As the properties of cirrus clouds may differ depending on the





geographical location (e.g., Heymsfield et al., 2017) and proximity to convection (e.g., Lawson et al., 2019), this study is associated with a deep convection case during Asian summer monsoon. Measurements in the TTL during F8 of the StratoClim campaign captured peaks of high ice water content up to 2400 ppmv and ice particle aggregates exceeding 700 µm in size with unusually long residence time. The ERA5 reanalysis indicates the presence of HIWC in the TTL but with an order of magnitude less than what was measured.

In this study, we used LES-scale simulations of a real case study to examine the sensitivity of TTL cirrus clouds to ice habit and microphysical parameterizations with predefined and non-predefined ice hydrometeor categories. Comparison of simulated microphysical properties to Stratoclim measurements shows that simulated HIWC in the TTL is sensitive to horizontal grid spacing, with HIWC increasing substantially as grid spacing decreases from 9 to 0.3 km, especially for the microphysics schemes lacking predefined ice categories. Notably, the microphysics scheme that predicts ice habit (Ishmael)

is the only scheme that reproduces the observed HIWC peak. The Ishmael scheme also simulates ice number concentrations comparable to the measurements. Particular attention is given to Ishmael, the microphysical parameterization predicting the ice habit. The corresponding fall speed and diabatic heating rates for each ice particle shape are compared between convective, thick anvil, and thin cirrus clouds. All simulations share the same vertical grid spacing of 100 m in the TTL.

The microphysical properties of cirrus clouds in the TTL from measurements and simulations are compared in the ice water

content-temperature (IWC-T) parameter space. Stratoclim measurements are dominated by ice particles of liquid origin with a small amount in the in-situ range. Simulated in-situ cirrus clouds are associated with low IWC and number concentration. Higher values of IWC are mainly of liquid origin and occur at colder temperatures. The simulated ice habits reveal that the ice particles of liquid origin, which substantially contribute to the HIWC, are of planar shape. The columnar and aggregates habits are mostly in the in-situ range with small IWC and number concentration. For all habits, ice number concentration increases

with decreasing temperature, which corroborates previous studies (e.g., Gryspeerdt et al., 2018; Sourdeval et al., 2018). The disparity in ice number concentration across ice habits can be up to four orders of magnitude, and aggregates are much less frequent than other habits. CPT and RHi, which regulate cirrus clouds, vary across schemes, and Ishmael scheme simulates them adequately.

The Ishmael scheme can produce the observed bimodal distribution of ice particle sizes. The lower range of the distribution is

mainly dominated by nucleation and diffusional growth, while the higher range is dominated by aggregation processes and depositional growth. The simulated bimodality is shifted towards larger values compared to the measurement. This study reveals that the microphysical evolution of cirrus clouds is strongly influenced by the type of ice habit in the cloud. In the TTL, the simulated ice habits evolve differently. The frequency of small planar ice particles increases faster due to enhanced in-situ nucleation. High RHi values above 130% with fall speeds below 10 cm s$^{-1}$ are only associated with the planar ice habit, leading

to an extended life cycle of cirrus clouds. For the planar ice habit, RHi is found to be inversely correlated with fall speed. This correlation is less evident for the other two ice habits. Columnar ice particles sediment rapidly without significant regrowth of aged ice particles, especially in the lower range of supersaturation with respect to ice. As these ice particles sediment, they may contribute to aggregate formation. Simulated aggregates exhibit different characteristics compared to pristine crystals.





The large aggregates with sizes between 600 and 800 µm have fall speeds of less than 20 cm s$^{-1}$, which corroborates the long
residence time of the aggregates observed during F8. We demonstrated and quantified the linear relationship between differential sedimentation of pristine ice crystals and the maximum dimension of aggregates. We found that the slope of this relationship depends on which pristine ice habit sediments faster. Therefore, ice habit could have a significant impact on the cirrus life cycle and the (de)Hydration of the TTL by regulating the sedimentation rate and the freeze-drying process. Further analysis revealed that ice habit also influences in-cloud radiative and latent heating rates, which are known to influence cloud
dynamics at a variety of scales (Gasparini et al, 2022; Ramaswamy and Ramanathan, 1989; Sherwood et al., 1994) scales. This demonstrates the importance of ice habit in shaping the properties and evolution of tropical cirrus clouds.

## 6 Code availability

WRF is available from the University Corporation for Atmospheric Research following the instructions at https://www2.mmm.ucar.edu/wrf/users/download/get_sources_new.php (University Corporation for Atmospheric
Research, 2022).

## 7 Data availability

All the data used in this study are openly available at Harvard Dataverse repository at
https://doi.org/10.7910/DVN/PKDDEI.

## 8 Author contributions

FL, MK and ZK conceptualized the study. FL ran the model and postprocessed the data. MK, AA and SK provided measurements from the campaign, details about instruments onboard of aircraft, and contributed to the interpretation. FL performed the analysis and writing of the original draft manuscript. FL, MK, AA, ABS, SK, AP and ZK edited the manuscript and provided feedback on the interpretation of the results.

## 9 Competing interests

Martina Krämer is a member of the editorial board of *Atmospheric Chemistry and Physics*. The remaining authors declare that they have no conflicts of interest.



## 10 Special issue statement

This article is part of the special issue "StratoClim stratospheric and upper tropospheric processes for better climate predictions (ACP/AMT inter-journal SI)".

## 11 Acknowledgments

FL and ZK are supported by OISE-1743753 and AGS-1759255. ABS is supported by NASA FINESST Grant 80NSSC20K1613. MK and AA are supported by the Seventh Framework Programme (SRATOCLIM) grant no. 603557. SK was supported by the Agence Nationale de la Recherche TTL-Xing project (ANR-17-CE01-0015).

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





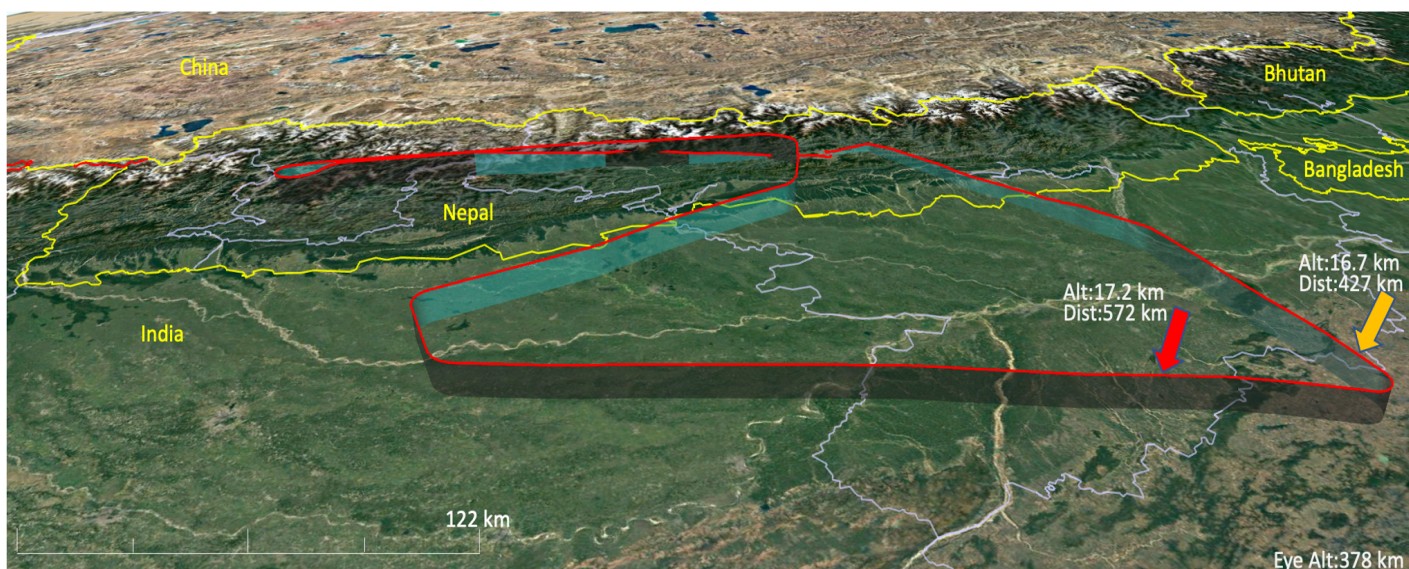

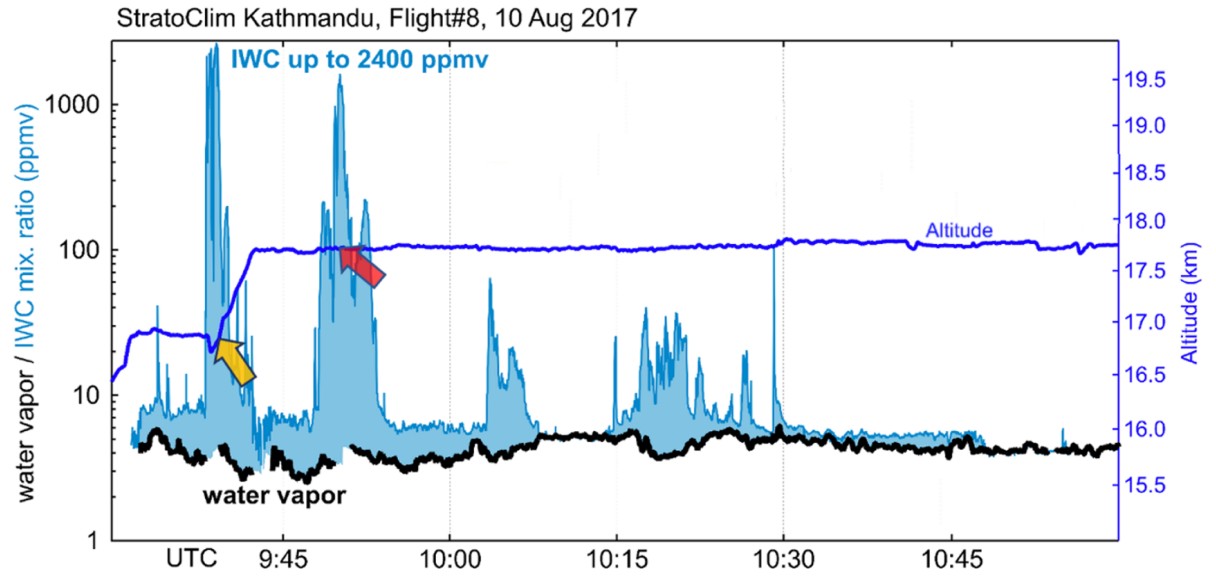

**Figure 1: Trajectory of Flight 8 (upper) and the corresponding time series of water vapor (black), ice water content (light blue), altitude (dark blue). Yellow and red arrows in both subfigures indicate high ice water content peak's location (upper) and timing (lower). The map was made using ©Google Earth 2022.**





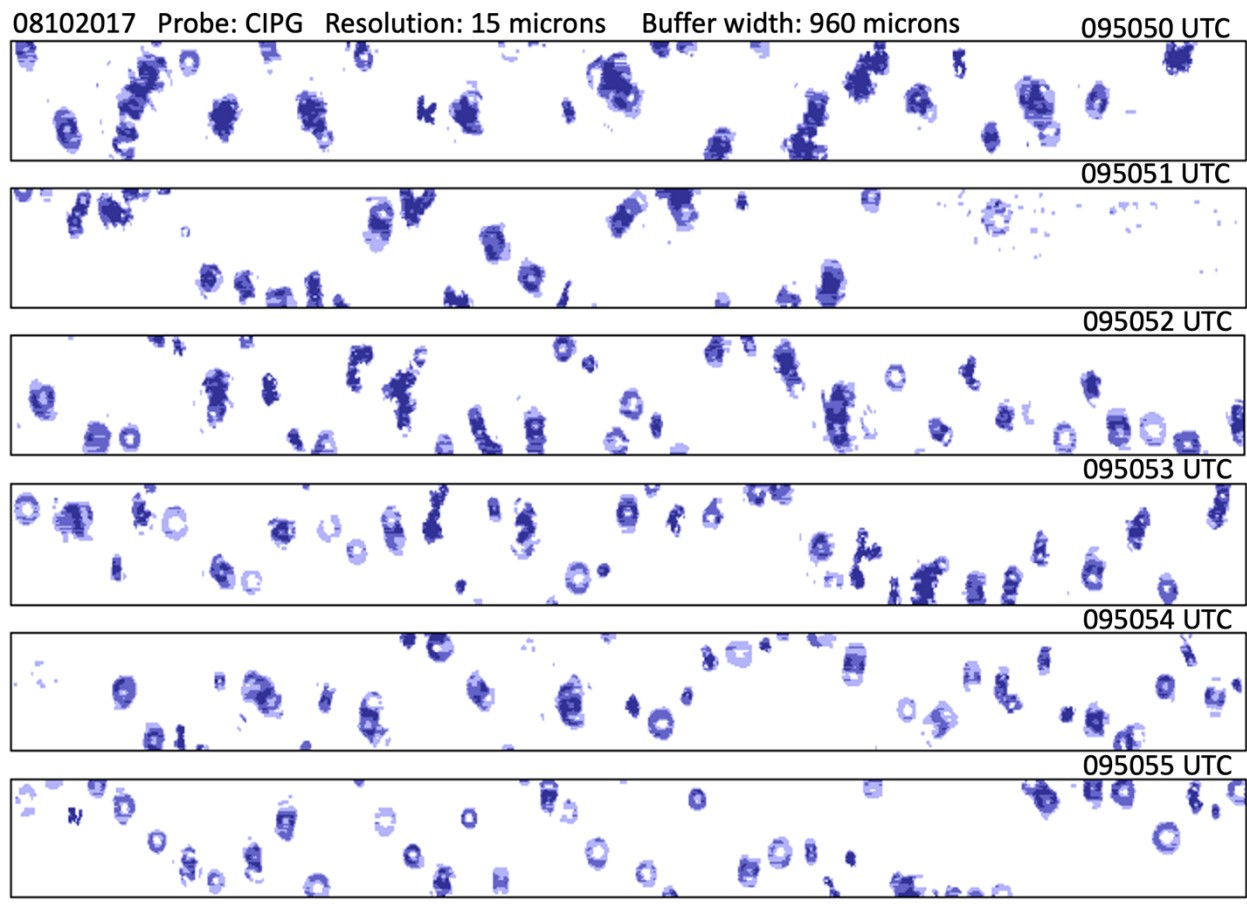

**Figure 2: Snapshots of ice particles images from CIPG probe during Flight 8 (F8).**








**Figure 3: WRF 4 nested domains (orange), Flight 8 trajectory (red) and closest CALIPSO passage (cyan). The map was made using ©Google Earth 2022.**



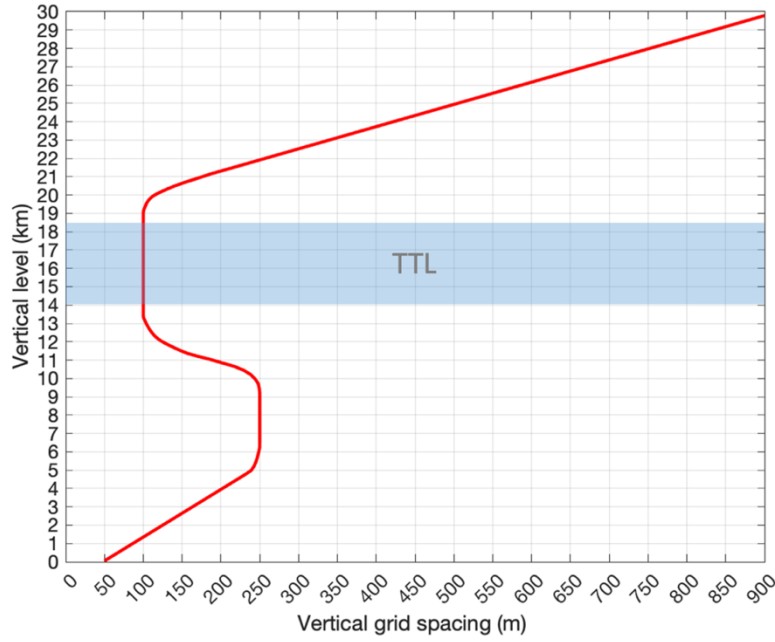


**Figure 4: Vertical spatial resolution for all simulations.**

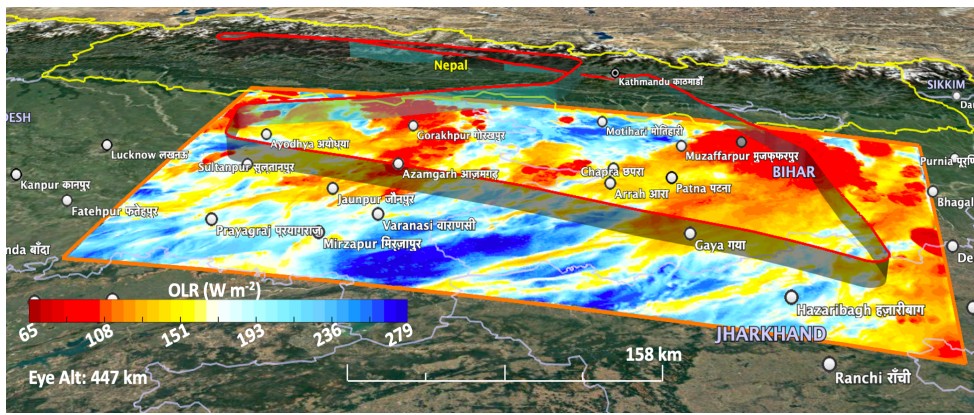

**Figure 5: High-resolution (0.3 km) simulated OLR on 1000 UTC 10 Aug 2017. The map was made using ©Google Earth 2022.**



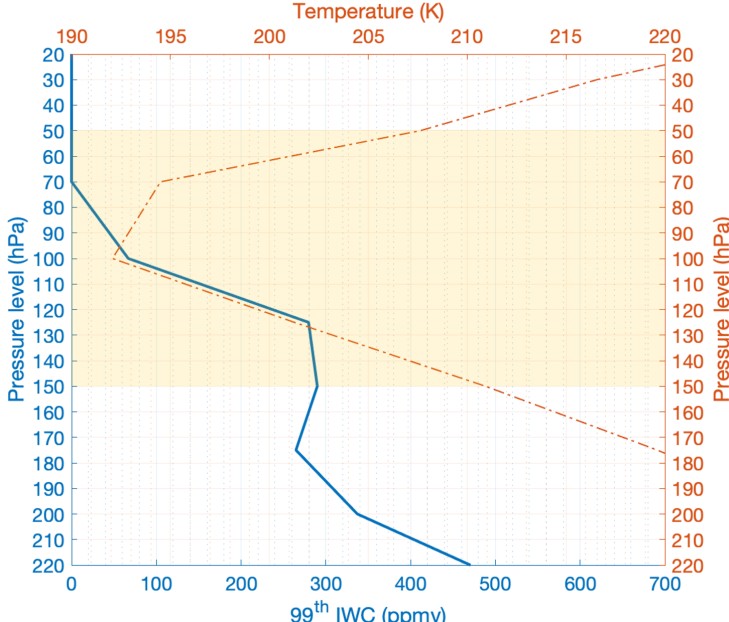

**Figure 6: Profiles below and within the TTL (yellow shade) in the inner-domain D4 from ERA5 99th of ice water content (blue solid line) and domain-mean temperature (red dash-dotted line), on 1000 UTC 10 Aug 2017.**



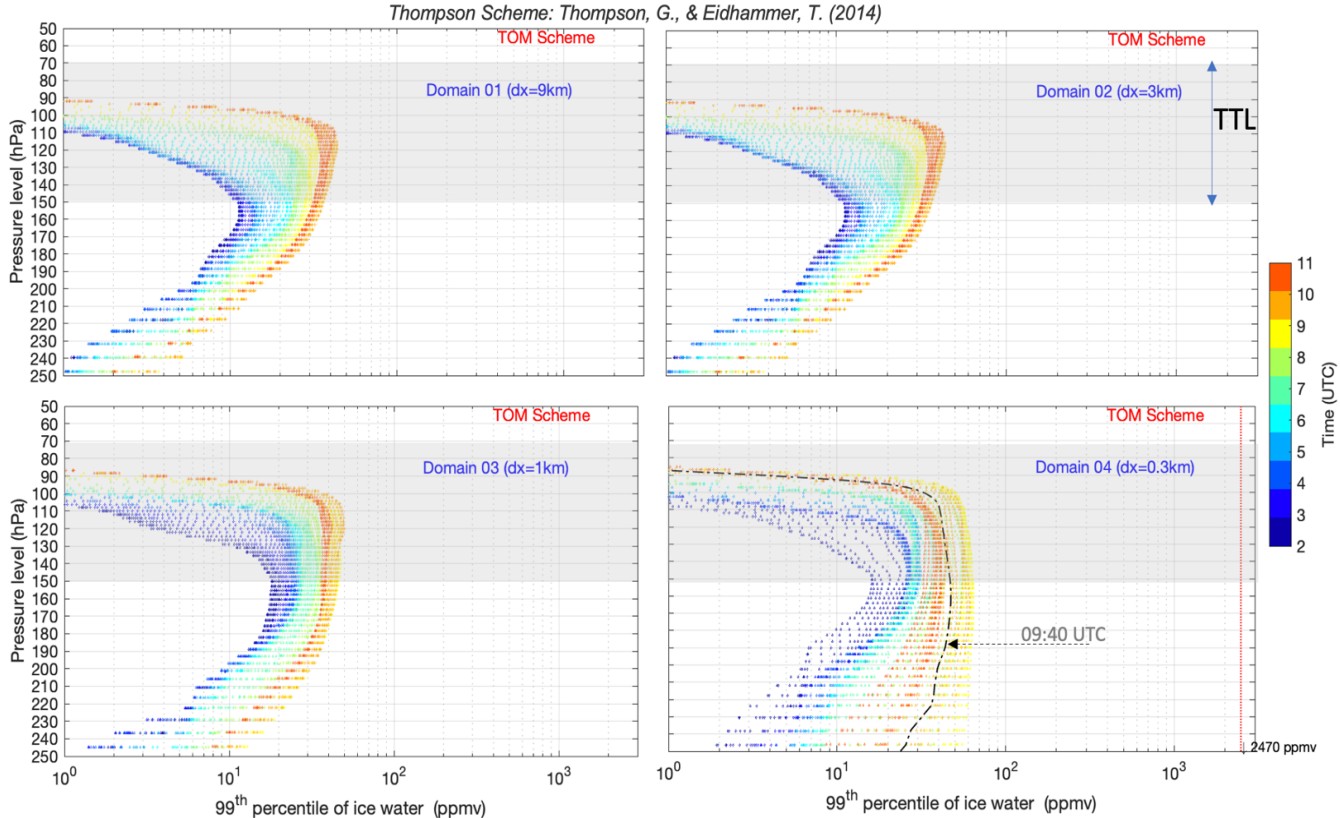

**Figure 7: Profile of 99th percentile of simulated IWC below and within the TTL (gray shade) using Thompson microphysics scheme, grid spacing of 9 km (upper left), 3 km (upper right), 1km (lower left) and 0.3 km (lower right) color coded by time (UTC) from 2 to 11 AM. Black dash-dotted line (lower right) represents the simulated profile at 09:40 UTC (timing of the first captured peak of IWC by aircraft) and the red vertical dotted line represents the observed value.**

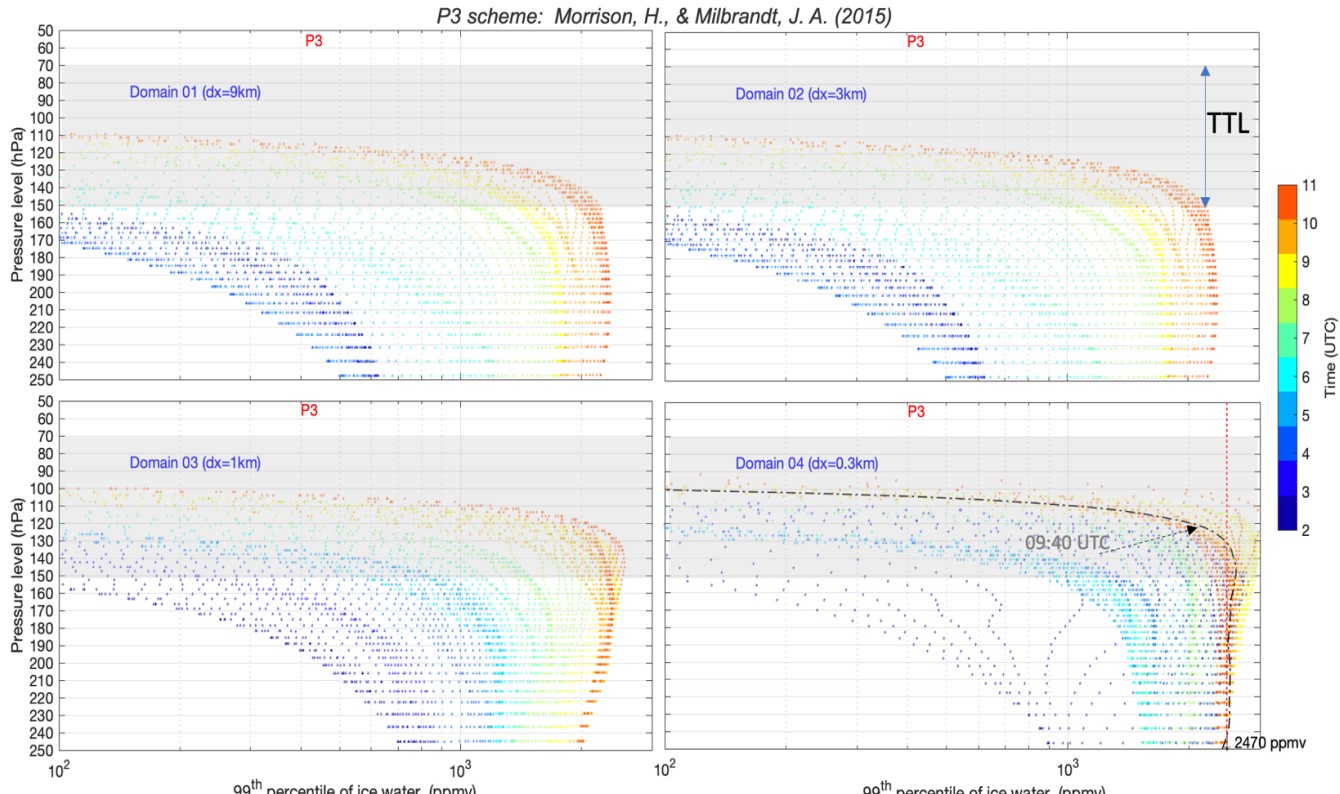

**Figure 8: Profile of 99<sup>th</sup> percentile of simulated IWC below and within the TTL (gray shade) using P3 microphysics scheme, grid spacing of 9 km (upper left), 3 km (upper right), 1km (lower left) and 0.3 km (lower right) color coded by time (UTC) from 2 to 11 AM. Black dash-dotted line (lower right) represents the simulated profile at 09:40 UTC (timing of the first captured peak of IWC by aircraft) and the red vertical dotted line represents the observed value.**




**Figure 9: Profile of 99th percentile of simulated IWC below and within the TTL (gray shade) using Ishmael (Jensen) microphysics scheme, grid spacing of 9 km (upper left), 3 km (upper right), 1km (lower left) and 0.3 km (lower right) color coded by time (UTC) from 2 to 11 AM. Black dash-dotted line (lower right) represents the simulated profile at 09:40 UTC (timing of the first captured peak of IWC by aircraft) and the red vertical dotted line represents the observed value.**





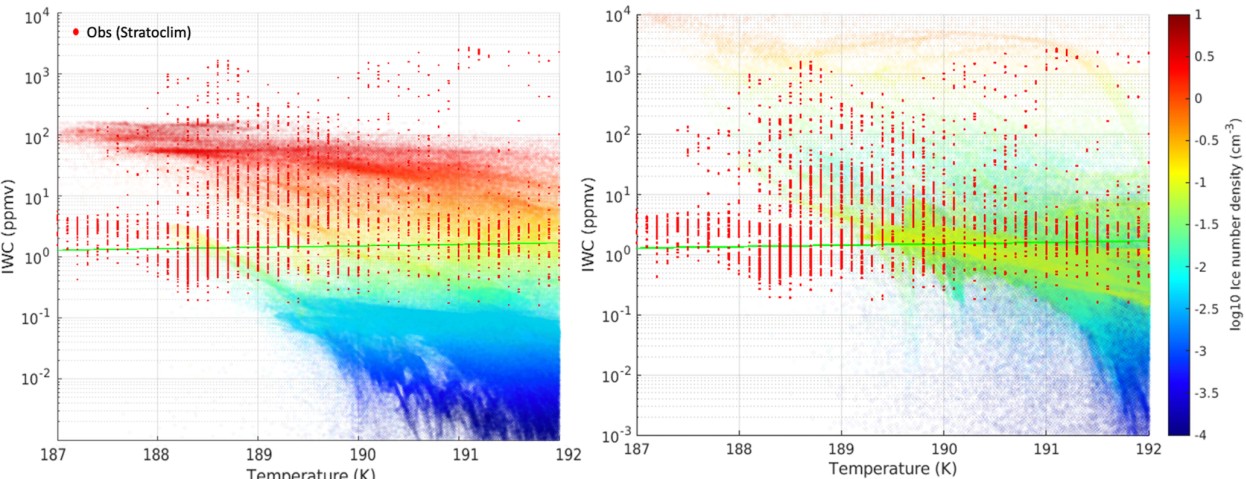

**Figure 10: Simulated (shaded) and observed (red dots) IWC versus Temperature color-coded by log$_{10}$ of ice number concentration at 17 km and for D_ice>3μm using Thompson (left) and P3 (right) schemes.**

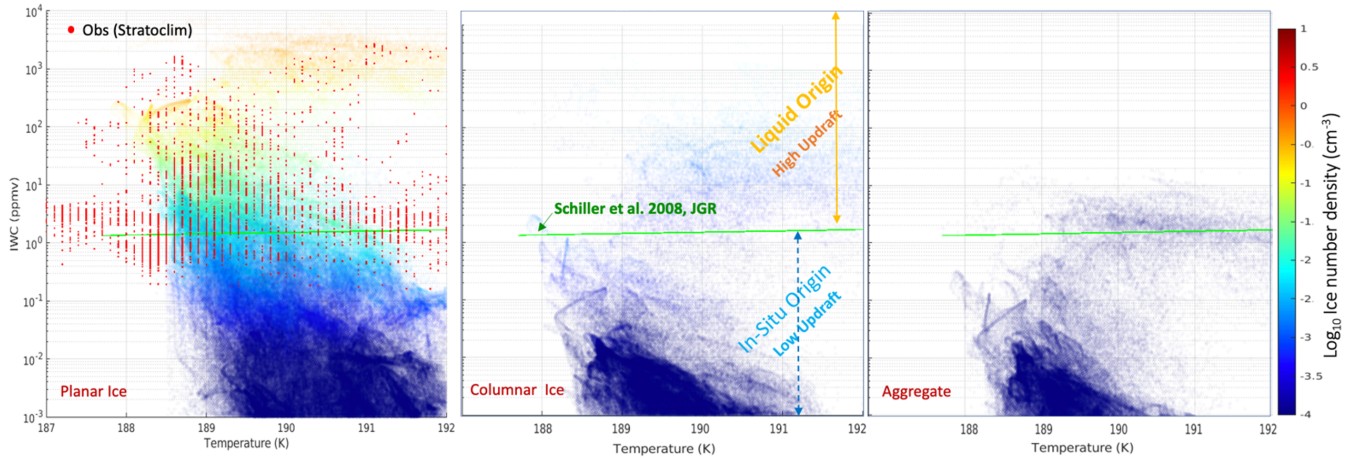


**Figure 11: Simulated (shaded) and observed (red dots) IWC versus Temperature color-coded by log$_{10}$ of ice number concentration at 17 km and D_ice>3μm using Ishmael scheme from three ice habits Planar (left), columnar (middle) and aggregates (right).**





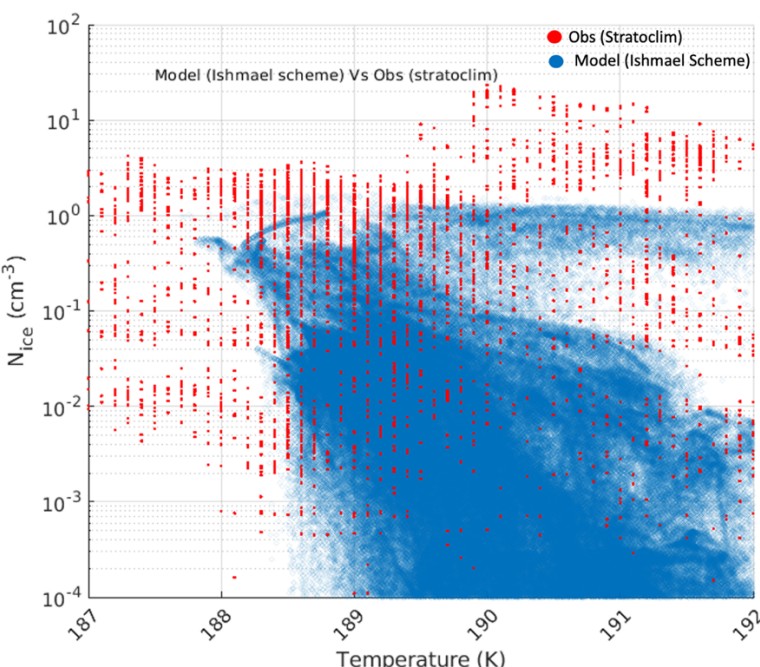

**Figure 12: Simulated (blue) and observed (red dots) $N_{ice}$ versus Temperature at 17 km using Ishmael scheme.**

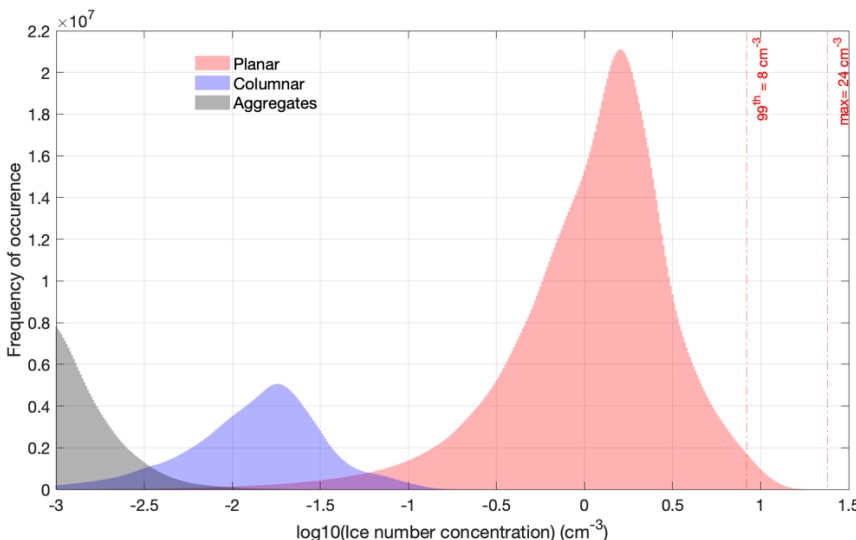

**Figure 13: Distribution of ice number concentration on a log10 scale using Ishmael scheme for different ice habits planar (red), columnar (blue) and aggregates (gray).**

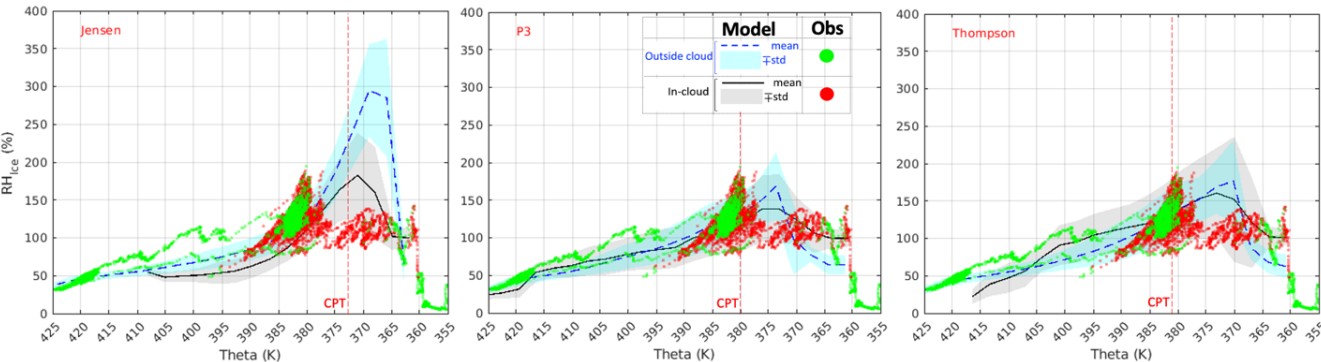

**Figure 14: Relative humidity with respect to ice versus potential temperature in-cloud and outside cloud from observation (red and green) and simulation (gray and blue) using Ishmael (left), P3 (middle) and Thompson (right) schemes.**

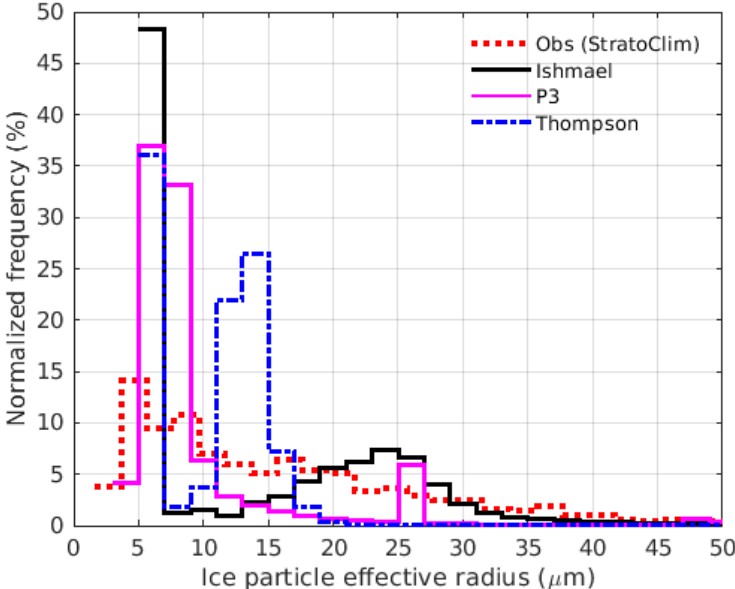

**Figure 15: Effective radius distribution at 17 km from observation (red) and simulation using Ishmael Jensen (Black), P3 (purple) and Thompson (blue) schemes.**



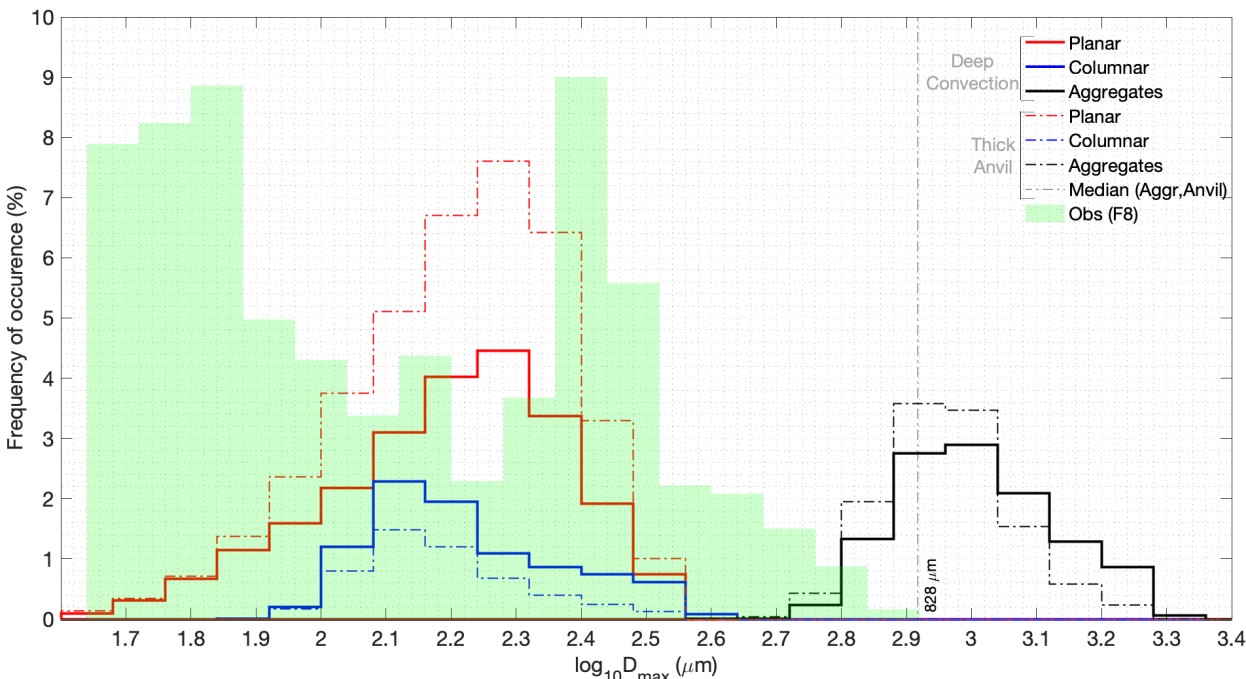

**Figure 16: Distribution of ice particles maximum dimension (D_max) from observation (green shade) and Ishmael scheme within the TTL in the deep convection (solid line) and anvil (dash-dotted) regions for ice habits; planar (red), columnar (blue) and aggregates (black).**

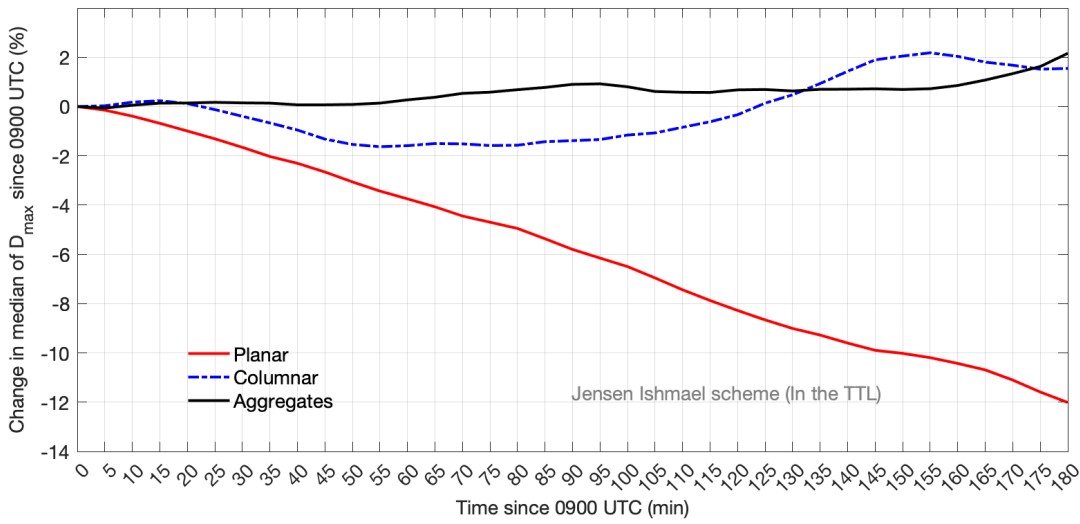


**Figure 17: Time series (starting from 2017-08-10 09:00 UTC) of the change in median value of simulated maximum dimension in the TTL using Ishmael scheme for ice habits; planar (red), columnar (dash-dotted blue) and aggregates (black).**



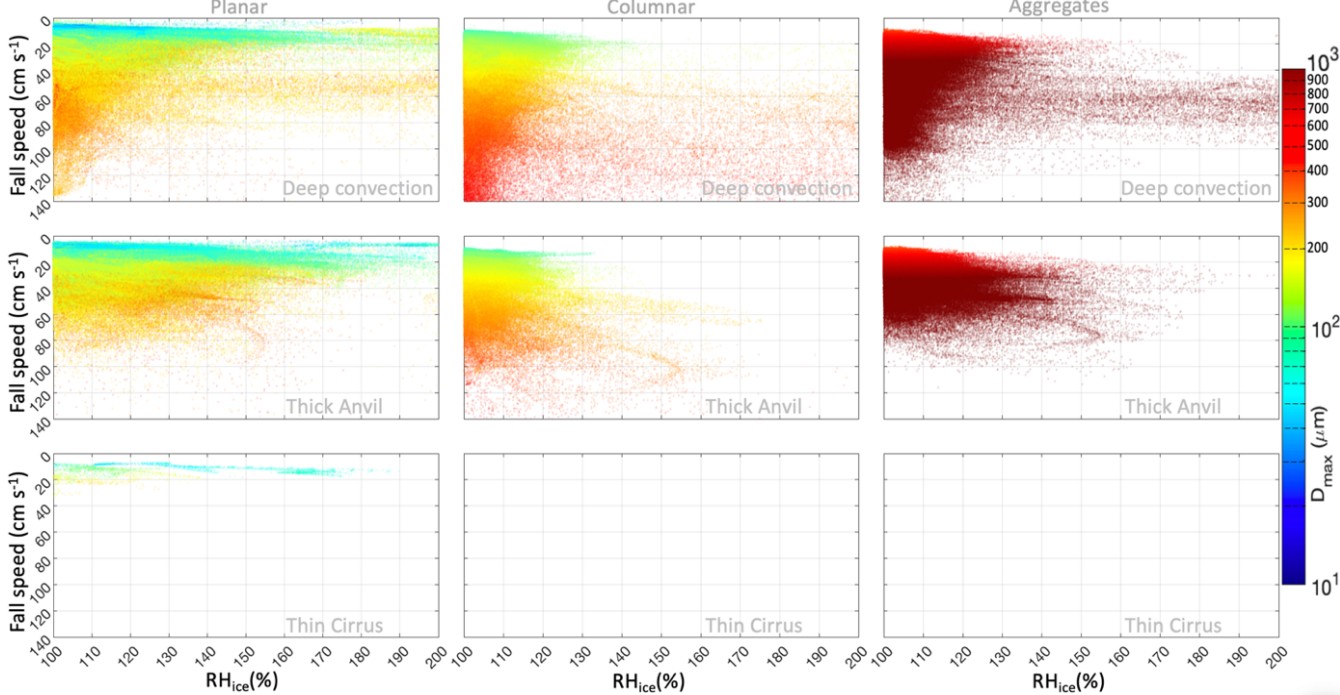

**Figure 18: Scatter plots of ice particle fall speed as a function of relative humidity with respect to ice (RHi), color coded by maximum dimension ($D_{max}$) for three different ice habits, planar (left column), columnar (middle column), aggregates (right column), and three different regions, deep convection (upper row), thick anvil (middle row) and thin cirrus (bottom row).**





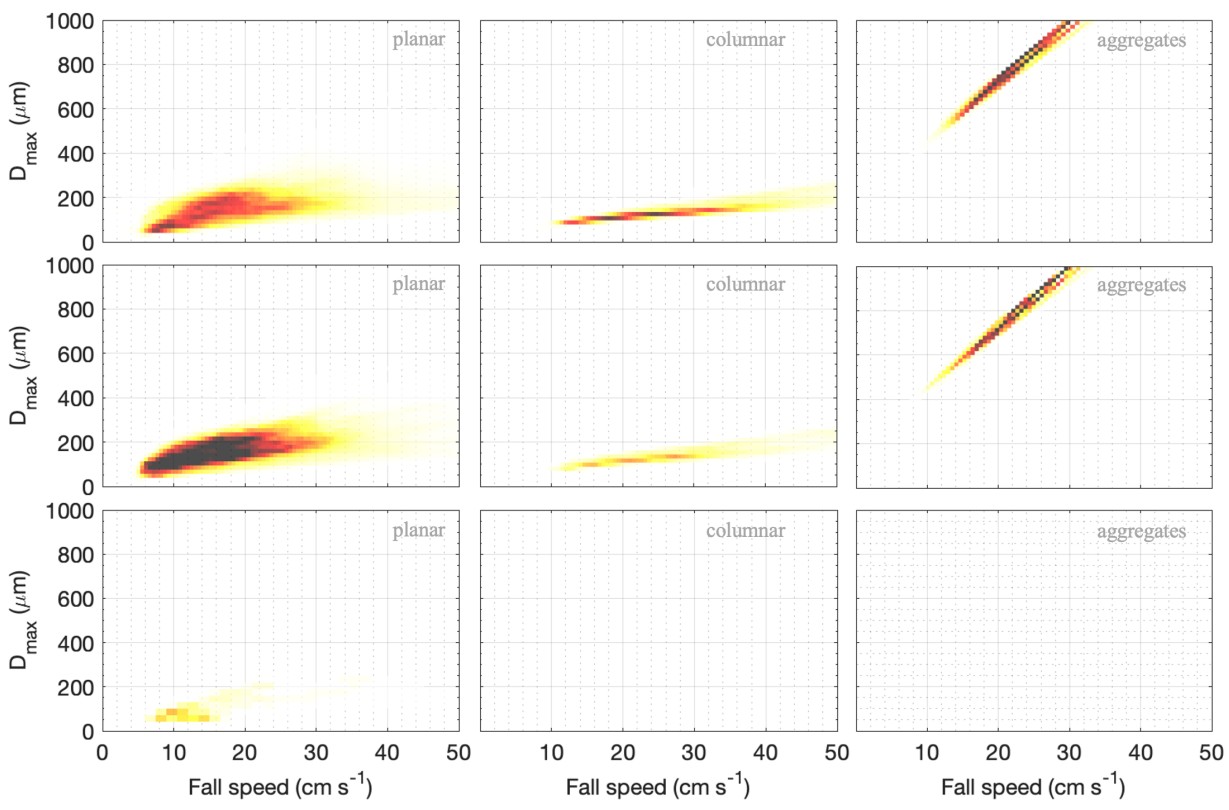

**Figure 19: Density plots of maximum dimension as function of fall speed, color coded by bi counts for three different ice habits, planar (left column), columnar (middle column), aggregates (right column), and three different regions, deep convection (upper row), thick anvil (middle row) and thin cirrus (bottom row).**

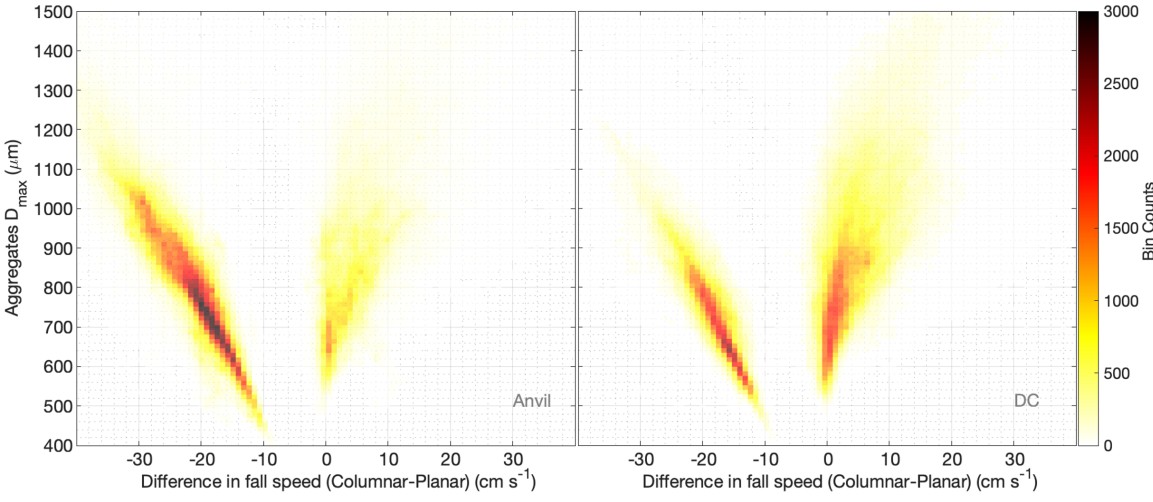

**Figure 20: Density plots of maximum dimension of aggregates as function of the difference in fall speeds of columnar and planer habits, color coded by bin counts for two regions in the TTL, deep convection (right) and thick anvil (left).**





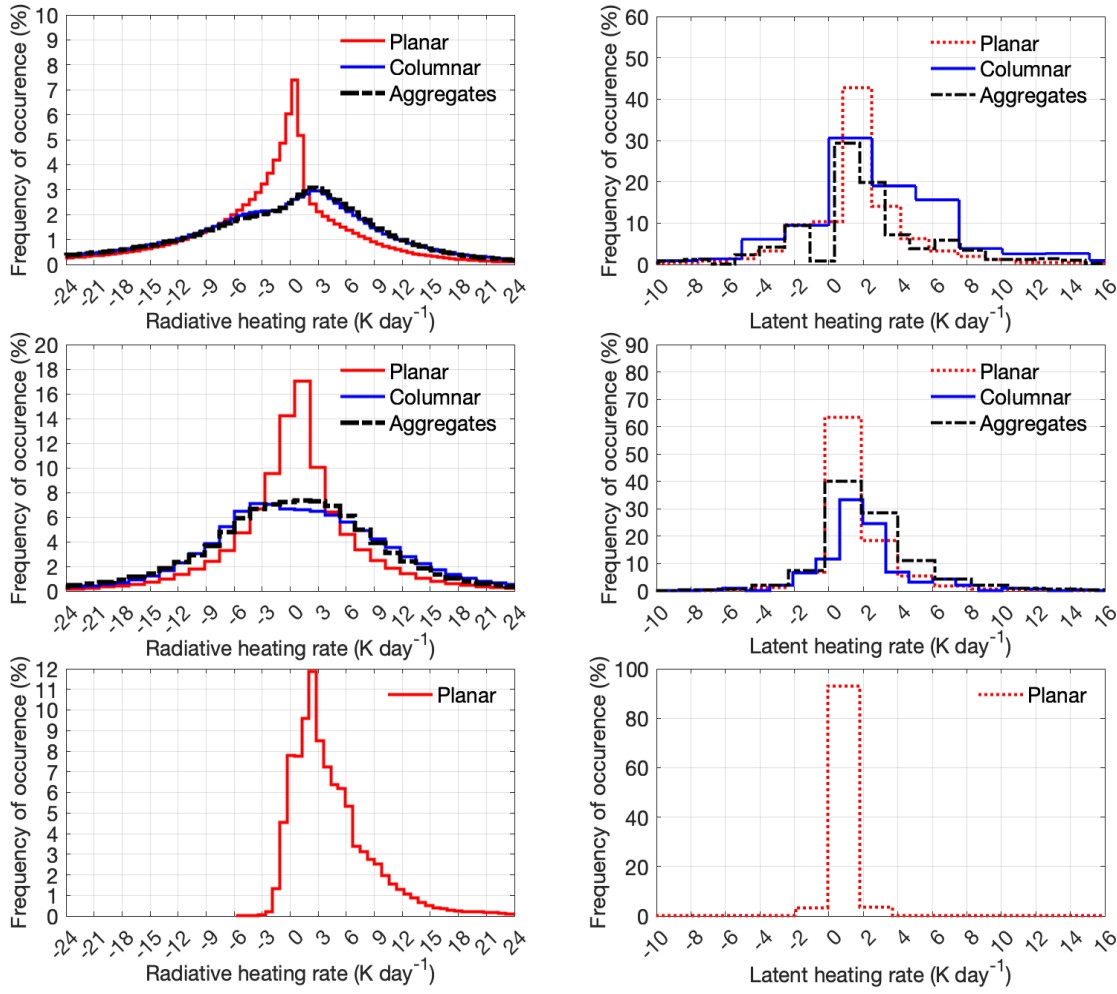

**Figure 21: Histograms of diabatic heating in the TTL from microphysics (right column) and radiation (left column) in the deep convection (upper row), thick anvil (middle row), and thin cirrus (bottom row) regions, and different ice habits; columnar (blue) planar (red) and aggregates (black).**
