# Peer review of "Sensitivity of convectively driven tropical tropopause cirrus properties to ice habit in high-resolution simulations"

_Atmospheric Chemistry and Physics, 2022_

## Author Comment (AC1)

**Response to comments from Anonymous Referee #1**

We would like to thank the reviewer for the constructive suggestions, which helped us improve the clarity of the paper. Based on the comments and suggestions, we have revised the manuscript.
Please find below our point-by-point responses (in blue) to the reviewer.
Thank you again for your time in assessing our work.

**Comment of Referee #1**
In this paper the authors use data from a measurement campaign and try to simulate a special case with three different microphysical schemes. They find that one of these schemes is better than the other two in reproducing the observations. This scheme is then further used to investigate a number of relations and dependencies between various quantities. The clouds are divided in three regimes and three forms of crystal habit are distinguished. The results are thus a bit complex and not easy to keep in mind.
To my view, this paper needs a major revision for the following reasons:
The presentation of the goals, methods and results is weak. I found many places where a rewording is necessary.  This is the major weakness, but it leads to unclarity in other respects.
For instance, although it is indicated in the abstract that the main goal of the paper is to judge the ability of the three codes to reproduce the observations, that is that this is a paper about models, this impression is not corroborated later, where the impression is, the main goal is the investigation of the role of the different cloud regimes and ice habits in regulating the properties of the TTL.

**Response to general comment of Referee #1**

The Abstract is modified and reworded to include:

1)      Clarification of the special issues and the motivation:
The lack of realistic representation of natural ice clouds habit in microphysical parameterizations and its impact on uncertainties in cloud-related processes and cloud-climate feedbacks
2)      Clarification of the goal: The investigation of the role of the different cloud regimes and ice habits in regulating the properties of the TTL.
3)      Reorganize the abstract to present how to treat the issue through the comparison of aircraft measurements with the simulations, presentation of observation and then the results that enable reaching out goal.

The reader would get more out of the paper if the findings could be presented in a more concise fashion, e.g. as a bullet list at the end. If you find a possibility to display your relations and sensitivities in a way that makes them physically plausible, it will even be better.

As suggested by the reviewer, the findings are presented in a form of bullet list in the summary.

Another important issue is the statement (several occasions) that the observed large crystals do not fall. I have no idea how this could be observed with an airborne measurement. Instead, it seems that clouds observed at different locations and times have been compared to simulated clouds at one location at different times. It might be that the crystals do hardly fall in the model, because of very low density and mass, but I find it questionable to conclude from this model result to the reality which was observed in a different setting.

We thank the reviewer for this comment regarding the longer residence time of large aggregates from observation.
Indeed, there is no way from the available aircraft measurements that confirm this claim. We draw our conclusion from the simulation only. The simulation indicated the presence of large aggregates with very low density and low fall speed (Fig. 16c,f).
The manuscript has corrected occordingly and the following statement has been removed from the manuscript.
"large aggregates do not fall faster and persist longer.."

**Response to specific comment of Referee #1**

In the following I list some observations that I have made at reading, in the order of their occurrence.
1) The title is not good and I think, it is misleading. The abstract indicates that the paper is not about physics but rather about the performance of different ice cloud model versions in reproducing certain observations. The title should reflect this. Further it is unclear which sentitivity is meant: "Sensitivity of ... cirrus", what does this mean?

Original Title: Sensitivity of convectively driven tropical tropopause cirrus to ice habit

New Title: Sensitivity of convectively driven tropical tropopause layer cirrus properties to ice habit in high-resolution simulations.

Added words:  "properties" after "cirrus"
                    "in high-resolution simulations" after "ice habit"

2) I also suggest to reorganise the abstract. The chain of arguments and statements does not form a logically coherent sequence, to my view. It starts with a general statement, then immediately jumps to the goal of the investigation, which however has to do with model details, not with something one could derive from the initial statement. Then it jumps again to reporting sentences on a field campaign. Please try to find a more coherent way to present 1) the general background, 2) the specal problem, 3) how you treated it, and 4) what results you found. Also this text should reflect that you are investigating a certain detail in models rather than a physical phenomenon.

As the reviewer suggested the abstract has been reorganized to form logically coherent sequence:

1- After the general statement at the beginning, the issue that we want to tackle is clarified by adding the following sentence. "The lack of realistic representation of natural ice clouds habit in microphysical parameterizations can lead to uncertainties in cloud-related processes and cloud-climate feedbacks."

2- The goal has been reworded to include different cloud regimes and ice habit. The following sentence has been added to the abstract "The main goal is the investigation of the role of different cloud regimes and the associated ice habits in regulating the properties of the TTL".

3- To introduce our approach and how we treat the problem, the following sentence is shifted before mentioning details from the campaign. "We compare aircraft measurements to a set of numerical experiments at the scales of large-eddy simulations (LES) of the same case study that employ different microphysics schemes."

3) Language should be checked carefully.
Line 45: "into" should be "in".
Corrected

Sentence beginning in line 55: "The impact ... is an ongoing debate". This is ugly style. I would rather say, the impact of ... is topic of a debate, but it is not a debate itself.
Corrected

Line 63 ff: "making them ... one of the largest uncertainties". No! the cirrus clouds are not uncertainties, but they may be the source of uncertainties.
Corrected

Lines 88-89: "We also investigate the impact of different ice habits on fall speed, relative humidity over ice and diabatic heating". Please reformulate. There are several issues in this sentence. First, I would not say that ice habits have an impact on fall speed (because there would be no fall speed if there were no crystals) but rather that their fall speed depends on their habit. Second, it is not clear to me how ice habit can impact on relative humidity, perhaps the sentence becomes correct, if cause and effect are reversed, i.e., the RHi impacts on the shape crystals attain.

Corrected as suggested and the sentence is replaced by the following:

"For ice crystals, we also explore how their fall speed depends on their habit, how relative humidity over ice impacts on the shape ice crystals attain. "

Line 105: correct "number size distribution"

Corrected

Figure 2: please add a scale, so that one can estimate the size of the crystals.

Scale has been added => Axis labels has been added to the bottom slide.
Also, the information about the scale is given in the header.
More details about the scale have been added to the figure caption "Each slide has a height of 960 microns and length of 1 sec".

Line 129: is the model indeed compressible or does it rather use assumptions about compressible air?

The ARW dynamics solver integrates the compressible, nonhydrostatic Euler equations.

Line 137: 900 km? or 900 m?

Corrected to 900 m, thanks

Line 160: please include the word "simulated" befor TTL. Cirrus clouds are per se not sensitive to any parameterisations.

Corrected

Lines 174-175 "but the growth of ice particle by vapor deposition remains constrained by mass–dimensional (m–D) and fall speed–dimensional (vt–D) relationships". I don't understand this. I think that ice growth is proportional to supersaturation, i.e. depends on RHi, and that the prefactor depends on a number of other things like crystal shape, deposition coefficient etc. In my view the growth of the crystals is constrained by available supersaturation. Please explain, how the mentioned relationships interfere with this simple physics.

Indeed, supersaturation among other variables such as capacitance, vapor diffusion coefficient, thermal conduction, and ventilation contribute to regulating vapor-depositional ice growth. For complex ice particle shape (non-spherical) this growth evolves differently along the minor axis and the major axis of the ice particle. The ratio of the minor axis *to* major axis growth is aspect ratio dependent (Chen and Lamb,1994; Jensen et al. 2017). For our study, we want to highlight the morphological impact on ice particles growth. With light riming, ice growth either thickens (extends the minor axis) or fills in the gaps (increase the effective density) of the ice particles, which leads to fall speed increase with no increase in the maximum dimension; this process contrasts with m–D relationships for rimed ice, which relate mass and fall speed increases to maximum dimension increases. Also, using m–D relationships means that all ice particles have the same mass for a given diameter. Thus, all simulated ice particles evolve along the same trajectory in mass–size space (Jensen and Harrington 2015; Jensen et al. 2017).
A brief explanation is included in the manuscript "Light riming can expand the minor axis of ice crystals or fill in gaps in their shapes, leading to increased density and fall speed without affecting maximum dimension."

Figures 7-9: the observations are represented in these figures by a red line which extends over the whole indicated pressure range. As the Geophysica does not measure an IWC profile but just a value at a certain altitude, I suggest to replace the line with a red dot or a short red bar that can cover the two altitudes of the two HIWC measurements. The current version misleads to the interpretation that P3 performs better than Ishmael, but that seems not to match your interpretation.

As suggested, a red double arrow has been added to mark the observed value at the corresponding observation levels. Also, following the suggestions of reviewer#2, the new figure 7 has only the profiles from the inner domain D04 for the three schemes. The other profiles with coarser resolution have been shifted to the appendix.

Line 230 "are likely heterogeneously, or, less likely, homogeneously" what?

This is corrected and reworded following the suggestion of reviewer #2.

 "…in-situ cirrus clouds that likely originate from heterogeneous nucleation, and less likely form homogeneous nucleation."

Text discussing Figs 10 and 11: Measurements show a lot of events at T<288K and around the green line, but simulations don't. Instead, the Thompson scheme shows crystals at this low temperature at much higher IWC and with large ice number density; the P3 shows crystals with IWC at 104 ppmv at T<188; the Ishmael show no events at this low T. Please provide more discussion about these findings. What is plausible, what not. And please, use "low temperature", not "cold temperature".

Replaced "cold/colder temperature" by "low/lower temperature" throughout the manuscript

We want to clarify that we are not claiming that the simulation captured the observed HIWC at the same time and at the same level. For the simulation, we are looking at HIWC in the whole TTL and comparing it with the observed HIWC. Ishmael scheme produced approximately similar HIWC peak to the measurement at the same time but at lower level within the TTL.
The reason why Thompson and P3 schemes show events at T<188 K and not Ishmael scheme is related to the simulated CPT levels and RHi. Both Thompson and P3 schemes overestimate the level of CPT (see fig RHi vs theta) which enables more simulated ice particles to reach higher and colder levels (T<188).
These two schemes have lower supersaturation with respect to ice compared to Ishmael scheme. This lower supersaturation decelerates ice particles growth and prevents them from reaching sedimentation threshold which enables ice particles to reach higher and colder levels.
Ishmael scheme simulates lower CPT level and higher supersaturation with respect to ice. This leads to accelerated growth rates, sedimentation, and cold trap at lower levels.  The cruising level of aircraft corresponds approximately to the level of the observed CPT.

The following explanation has been added to the manuscript.
"Only The Thompson and P3 schemes indicate events at T<188 K which can be explained by the simulated CPT levels and RHi. The Thompson and P3 schemes overestimate the CPT level (Fig.10), allowing more simulated ice particles to reach higher and colder levels (T<188). Both schemes have lower supersaturation with respect to ice compared

to the Ishmael scheme. This lower supersaturation slows the growth of ice particles and prevents them from reaching the sedimentation threshold, thus allowing ice particles to reach higher and colder levels. The Ishmael scheme simulates a lower CPT level and higher supersaturation with respect to ice. This leads to accelerated growth rates, more sedimentation and cold trap at lower levels. The aircraft cruising level corresponds approximately to the observed CPT level."

Text discussing Fig 14:  To my view, this discussion can be improved. Currently there is just a description of the figures.
I am, for instance, much interested to know why the so far best scheme shows so high supersaturation values. Do you think, they are realistic?
What would happen, if there were indeed such high values of RHi, both in nature and in the model. Would you expect new nucleation (any mechanism)?
 Does the extreme supersaturation in the model explain the occurrence of the >800µm crystals?

To answer all the questions about Fig 14, the following details have been added to the manuscript with the text discussing Fig 14.

"According to previous laboratory studies (e.g., Bailey and Hallett, 2009), RHi can reach high values of ~ 160 % at very low temperatures. Also, high RHi values of ~200 % at the tropopause level were reported in previous field campaign studies (Krämer et al., 2020; Dekoutsidis et al. 2022) but with very low frequency of occurrence of ~2%.  Simulations and especially Ishmael scheme overestimate RHi. The reason that may explain the existence of such high supersaturations is the low ice number concentration, which is mainly present at low temperature (Fig. 8). Previous studies (eg., Jensen et al. 2013) have also indicated that increasing the number of ice crystals leads to decrease in RHi until reaching the saturation. The Thompson and P3 schemes produce smaller ice particles compared to Ishmael (Fig 11) and high ice number concentration (Fig 8). For this reason, the Ishmael scheme produces a higher RHi (Fig. 10). With high RHi and lower temperatures more in-situ nucleation, faster growth rate of crystals and larger ice particles are expected. The comparison between the three habits shown in Figure 9 indicate that large aggregates are associated mainly with low number concentration which demonstrate the correlation between aggregates and high supersaturation (Jensen et al. 2013). RHi values of 120% and 140% can be considered approximate thresholds for heterogeneous and homogeneous ice nucleation respectively (eg., Koop et al., 2000; Kärcher, 2012). Therefore, due to the high supersaturation with respect to ice, more homogeneous and in-situ nucleation is associated with Ishmael scheme than the other two schemes."

Dekoutsidis, G., Groß, S., Wirth, M., Krämer, M., and Rolf, C.: Characteristics of supersaturation in mid-latitude cirrus clouds and their adjacent cloud-free air, Atmos. Chem. Phys. Discuss., https://doi.org/10.5194/acp-2022-717, in review, 2022.

Jensen, E., Diskin, G., Lawson, P. Lance, S., Bui, T., Hlavka, D., McGill, M., Pfister, L., Toon, O., and Gao, R.: Ice nucleation and dehydration in the Tropical Tropopause Layer, P. Natl. Acad. Sci. USA, 110, 2041–2046, https://doi.org/10.1073/pnas.1217104110, 2013

Kärcher, B.: Supersaturation Fluctuations in Cirrus Clouds Driven by Colored Noise, J. Atmos. Sci., 69, 435–443, 2012.

Koop, T., Luo, B., Tsias, A., and Peter, T.: Water activity as the determinant for homogeneous ice nucleation in aqueous solutions, Nature, 406, 611–614, 2000.

Krämer, M., Rolf, C., Spelten, N., Afchine, A., Fahey, D., Jensen, E., Khaykin, S., Kuhn, T., Lawson, P., Lykov, A., Pan, L. L., Riese, M., Rollins, A., Stroh, F., Thornberry, T., Wolf, V., Woods, S., Spichtinger, P., Quaas, J., and Sourdeval, O.: A microphysics guide to cirrus – Part 2: Climatologies of clouds and humidity from observations, Atmos. Chem. Phys., 20, 12569–12608, https://doi.org/10.5194/acp-20-12569-2020, 2020.

And why should high RHi favour an enhanced freeze drying process? To me, freeze drying means, that ice crystals consume all the available supersaturation at the coldest layer, such that only ice saturated vapour (RHi=100%) enters the stratosphere. If we however have huge RHi in a vertical uplift, freeze drying would have to act very quickly to get supersaturation consumed completely. This sounds implausible. So to my view, freeze-drying is not favoured at all, it simply gets unimportant at such conditions.

The following claim (sentence) about high RHi favors freeze-drying has been removed from the manuscript
"..favors an enhanced freeze-drying process and further.. "

Line 266: how can a distribution be underestimated? Please reformulate.

Corrected to be "simulated effective radius distributions are within smaller size ranges of ice particles compared to the observations,.."

Fig 15: As an effective radius is somehow the ratio of the third to the second moment of the size distribution, I think, it should be a single value. But here you show distributions.
Do you consider the effective radii in every grid box, respectively, and then construct the histogram from these values? Please clarify. Why does the Ishmael scheme not produce larger crystals if the RHi is so large in this model? Is this because sedimentation removes such crystals quickly? And finally, why do you show effective radius instead of a more simple measure like volume equivalent radius or maximum dimension?

indeed, the values of the effective radius are single values. Clarification about the histogram has been added to the manuscript.
"The effective radii in every grid box are considered, respectively, and then the histogram is constructed from these values."

Ishmael scheme is dominated mainly by the planar habit with the highest frequency of occurrence. For this habit, large RHi favors in-situ nucleation with small crystal sizes and large crystals with high ice density that sediment rapidly. Large crystals that remain are mostly aggregates with low frequency of occurrence and very low ice density.

Not all schemes generate the maximum size. For the purpose of comparison across schemes, we have chosen to use the effective radius which is available for all schemes and observation. We use the maximum dimension with the Ishmael scheme which is the focus of this study with its ice habits.

Line 273: "lower RHi saturation" is sloppy and unclear. Saturation means RHi=100%. Or do you mean the corresponding vapour pressure?

Corrected and removed the word saturation "lower RHi"

Fig 16: It seems here that the model produces much larger crystals than observed, in line with the huge supersaturation. But in figure 15 we saw that the model has smaller effective radii than the observations. How can you explain that. It sounds like a contradiction.

It has to do with the low frequency and number concentration of simulated large aggregates. The number concentration and frequency of occurrence has significant impact on the calculation of the effective radius. The aggregates have very low frequency of occurrence and number concentration compared to columnar and planar habits => The effect of simulated large aggregates with their large values of maximum dimension is minor in the simulated effective radius. The ratio of simulated large aggregates to other habits is much smaller than the same ratio from observation. The simulated PSD is dominated by planar and columnar habits which is underestimated compared to observation.

Line 306 ff "During F8, it was reported that very large aggregates of ice particles were observed for an unusually extended long time without falling". How is this possible? Has the Geophysica made a stop there and somebody of its crew watched certain large ice crystals that were not falling?
Or do you rather mean, you found large ice crystals at one place and a while later you found again large ice crystals, but at another place, and later. This would not imply the implausible interpretation that these crystals were not falling. Please explain more exactly what you mean. And then rethink your interpretation. The Geophysica observes different clouds or cloud parts at different times (the flight path does not show circling somewhere), but the simulations represent one cloud at one place at different times. Obviously these are different settings. How can you then relate the two. This sounds illogical.

The sentence has been reworded to avoid ambiguity
"During F8, very large aggregates of ice particles were observed."

Line 326: Perhaps the sentence becomes better, if you include "and thus indirectly affects...".

"and thus indirectly affects" has added to the sentence as suggested

Line 381: Please rewrite the sentence in a way that the reader does not conclude that ice in the TTL (i.e. in the real atmosphere) is in any way dependent on details of any model simulations.

For clarification the word "simulated" has been added before "HIWC".

Line 381ff: please explain how the distributions have been obtained: are they taken over the whole model domain (each grid cell, but divided acc. to the cloud categories) and at each time step, or was the sampling different?

The following details have been added:
"The distributions of the two heating rates are taken over the entire domain at each grid cell every 5 minutes which is the output frequency of the simulations. The distributions are divided according to the cloud categories that are based on ice water path thresholds."

Section 4.5: I wonder what heating rate distributions for different habits and different cloud regimes mean in total, i.e. which regime and which habit has the biggest effect. To answer this question you would need to integrate the histograms and weight them with the respective frequencies of occurrence. Is that possible? It should tell us how many Gigajoules (or something similar) are stored in the three hours in the TTL, due to the presence of different crystal shapes in different cloud regimes. The mentioned doubling in heating rates could imply a considerable larger or smaller factor in the total energy deposited in three hours. I think, this is probably more relevant than the heating rate distributions.

As suggested, we calculated the energy stored in the three hours in the TTL. For the deep convection regime, we found -12.4, -5.1, -5.6 Gigajoules for Planar, columnar and aggregates habits respectively. For the thick anvil regime, we found -9.2, -1.5 and 0.7 Gigajoules for Planar, columnar and aggregates habits respectively. For Thin cirrus, we found 0.4 Gigajoules for Planar habit.
We would like to point out that the additional section on radiative heating is not the focus of this study. This section is added to show qualitatively that the shape of ice crystals can influence diabatic heating rates in TTL cirrus clouds. Furthermore, this section is added to motivate future studies to investigate further and provide an in-depth analysis of the impact of ice habit on diabatic heating rates in TTL cirrus clouds.

---

## Author Comment (AC2)

**Response to comments from Anonymous Referee #2**

We would like to thank the reviewer for the constructive suggestions, which helped us improve the clarity of the paper. Based on the comments and suggestions, we have revised the manuscript.
Please find below our point-by-point responses (in red) to the reviewer.
Thank you again for your time in assessing our work.

Line 46: Can the authors clarify the sentence starting with "Therefore, air injected into the lower stratosphere…."? Based on the writing I am not sure how the previous sentences result in this statement. Also, "than into the troposphere" should be "than in the troposphere".
"In the tropics, the transport by the Brewer-Dobson circulation is upward. Therefore, air injected into the lower stratosphere has a longer residence time than into the troposphere and reaches further distances into the stratosphere."

For clarification, the sentence is reworded and replaced by the following:

"In the upwelling branch of the Brewer-Dobson circulation (BDC), the water vapor injected in the tropical lower stratosphere is transported meridionally to the temperate stratosphere. Due to the slow nature of BDC, this process results in a longer residence time of the injected stratospheric water vapor."

Line 48: is there a reference to explain the "tropically averaged level of all-sky zero net radiative heating"?

Added the following references for "zero net radiative heating" :

(Gettelman and Forster, 2002; Sherwood and Dessler, 2001,2002).

Gettelman, A., and P. M. de F. Forster, 2002: A climatology of the tropical tropopause layer. J. Meteor. Soc. Japan, 80, 911–924.
Sherwood, S. C., and A. E. Dessler, 2000: On the control of stratospheric humidity, Geophys. Res. Lett., 27, 2513 – 2516.
Sherwood, S. C., and A. E. Dessler, 2001: A model for transport across the tropical tropopause, J. Atmos. Sci., 58, 765 – 779.

Line 61: please clarify "cold trap". Is this the CPT? Also, delete "On the other hand" at the beginning of the sentence.

"On the other hand" → has been deleted

The cold trap is the coldest region in the tropical tropopause layer (TTL) that modulates the water vapor entering the lower stratosphere via cirrus formation (Jensen et al., 1996) and ice crystals sedimentation (Holton and Gettelman 2001). The corresponding low water vapor concentration and freeze-dried air is indicative of the dehydration process (Ryu and Lee, 2010).

Such freeze drying would appear to require slow upwelling in that region to produce cooling and condensation and to allow time for the ice crystals formed in the freezing process to sediment out.

The following details about the cold trap has been added to the manuscript.

"…Ice particles injected into the lower stratosphere bypass the cold trap which is the coldest region in the TTL that modulates the water vapor entering the lower stratosphere via cirrus formation (Jensen et al., 1996) and ice crystals sedimentation (Holton and Gettelman 2001), sublimate above the level of CPT, and contribute to stratospheric hydration.."

Jensen, E. J., O. B. Toon, H. B. Selkirk, J. D. Spinhirne, and M. R. Schoeberl, On the formation and persistence of subvisible cirrus clouds near the tropical tropopause, J. Geophys. Res., 101, 21,361- 21,375, 1996.

Ryu, J., & Lee, S. (2010). Effect of Tropical Waves on the Tropical Tropopause Transition Layer Upwelling, Journal of the Atmospheric Sciences, 67(10), 3130-3148.

Line 63: This should be the start of a new paragraph. I can't tell if that is the authors' intention in this template. If so, then ignore this comment.

Corrected

Line 66: the authors could expand this to include other estimates. Satellite based estimates by Sassen and L'Ecuyer place cirrus coverage much lower than what is quoted here.

To add the suggested references, the following details have been added to the manuscript:
"Spaceborne active remote sensing (i.e., CloudSat and Calipso) reported 35% of cirrus coverage within ±15° latitude and 56% within ±30° latitude of the equator (*Sassen et al.*, 2008, 2009). According Baran et al (2012), satellite measurements in the tropics revealed that cirrus cloud cover can be 60–80% at any given time."

Sassen, K., Z. Wang, and D. Liu (2008), Global distribution of cirrus clouds from CloudSat/Cloud-Aerosol Lidar and Infrared Pathfinder Satellite Observations (CALIPSO) measurements, J. Geophys. Res., 113, D00A12, doi:10.1029/2008JD009972.

Sassen, K., Z. Wang, and D. Liu (2009), Cirrus clouds and deep convection in the tropics: Insights from CALIPSO and CloudSat, J. Geophys. Res., 114, D00H06, doi:10.1029/2009JD011916.

Line 68: GCMs acronym is never defined

GCMs acronym is defined:   (General Circulation Models)

Line 130: "(Fig. 1)" should refer to Fig. 3 instead.

Corrected: "Fig. 1" → "Fig. 3"

Line 169: delete "problematic" and describe this more clearly why artificial conversion
rates are unphysical

", which is inherently problematic" → removed

The artificial conversion between particle types is not physical because it is arbitrary and relies on ad-hoc
tunable parameters that cannot be directly measured. It leads to systematic bias of the model output.

Added → : " which relies on ad-hoc tunable parameters that cannot be directly measured".
Also, more details are stated in line 176

"Artificial transitions between predefined ice categories and the corresponding ad-hoc conversion
parameters can systematically bias the model output (Morrison and Grabowski, 2008; Lin and Colle,
2009)."

Line 175: why do the m-D and vt-D relationships not properly capture the transitions in
ice particle mass and fall speed during riming?

The explanation is in the following sentences.

Line 181 :"With the use of m–D and vt–D relationships, it is assumed that all modeled ice particles evolve
and sediment along the same trajectory in mass–size and fall speed-size spaces."

Line 191:
"Light riming can expand the minor axis of ice crystals or fill in gaps in their shapes, leading to increased
density and fall speed without affecting maximum dimension."

Light riming extends the minor axis of ice particles or fills in gaps to increase the effective density of ice
particles, resulting in an increase of fall speed without an increase in maximum dimension. This process
contrasts with the m–D relationships for rime ice, which relate increases in mass and fall speed to
increases in maximum dimension (Jensen et al., 2017). The shapes and fall speeds of ice particles are
assumed a priori.

Line 198: please explain what is meant by the 99th percentile and how this is calculated.
Also, can the authors explain why they chose the 99th percentile? Why not the 75th
percentile?

We consider $99^{th}$ percentiles (top 1%) of Ice water content to mainly reflect high ice water content (HIWC).
The $75^{th}$ percentile is not representative for this extreme value.

Line 202: add "value" to the end of the sentence

Added

Line 205: The statement beginning with "to examine the sensitivity…" includes a lot of repeated information from the methods, please reword.

Repeated details have been removed
"To examine the sensitivity of simulated HIWC to changes in model grid spacing and microphysical parameterizations, we performed a set of simulations configured with four nested domains and three microphysics schemes."

Line 229: please reword "Below this line, ice particles are associated with weak updrafts and represent mostly in-situ cirrus clouds that are likely heterogeneously, or, less likely, homogeneously.". The latter half of the statement should read something like, "…in-situ cirrus clouds that likely originate from heterogeneous nucleation, and less likely form homogeneous nucleation."

Corrected as suggested

Line 246: this is not clear based on Figure 12

Corrected to (Fig.9a, left).

Line 266: again, the 99th percentile should be defined somewhere in the text, in line with my previous comment. The figure does not show the 99th percentile of the data for the measurements or any of the simulations except for the Thompson scheme. I would also say that all simulations overestimate smaller ice crystals compared to the observations. See comments on Figure 15 below. Finally, why does the Thompson scheme stand out from the others between 10-15 um.

The 99th percentile is defined is line 204

"The 99$^{th}$ percentile is used to present HIWC which refers the top 1% of simulated or observed IWC values."

The 99$^{th}$ percentile (top 1%) is calculated for all schemes but not shown in figure 15.
Indeed, all simulation overestimate smaller ice crystals compared to the observation.

The following sentence has been added in lines 305-307: "…, especially for Thompson. It can be explained by the slightly warmer CPT (Fig. 12) compared to the other two schemes which can limit ice crystal growth (Fig. 13) and LWC (Figs. 7 and 8)."

Line 315: please add a reference to figure 14 after "due to the difference in RHi"

Old figure 14 became figure 12.

reference to figure 12 has been added.

Line 391 (319): isn't the statement "the low rate of in-situ nucleation" contradictory of the statement above saying that the PSD indicates more in-situ nucleation of planar ice crystals in the anvil?

There is no contradiction: The planar habit experiences higher in-situ nucleation but strong decrease in ice crystals with large maximum dimension due to sedimentation.
The lower rate of in-situ nucleation is for columnar habit.  For clarification, the sentence is reworded as follows in lines 353:354

"For columnar habit, the low in-situ nucleation rate prevents the ice crystals from reaching the sedimentation threshold and from undergoing a strong decrease in the number of large ice crystals encountered by the planar habit (Fig. 15)."

Line 330: shouldn't "scatter plots of RHi as a function of fall speed" be "fall speed as a function of RHi"?
Corrected as suggested.

Also, it is unclear why this connection is being made. These are seemingly unrelated quantities. Can the authors add some explanation perhaps?

The RHi regulates the nucleation and growth rates of ice particles. The shape of ice crystals can change significantly over a few degrees Celsius (e.g., Magono and Lee 1966; Bailey and Hallett 2009) and influences its fall speed in the cloud (Heymsfield and Iaquinta 2000).
Freshly nucleated ice crystals of small sizes and low fall speeds can be either in situ or of liquid origin. The figure indicates a wide range of RHi for the same fall speed, especially for slower falling ice particles. For small ice particles (blue color) and low fall speed (<20 cm s$^{-1}$), the associated RHi values help distinguish in-situ and homogeneous nucleation with high RHi from liquid origin with lower RHi.
Moreover, this figure reveals that planar habit dominates in-situ nucleation, especially in the anvil.
Therefore, this figure relates these three variables to help identify the origin of the nucleation process, its region in the cloud, and the associated habit.

Heymsfield, A. J., and J. Iaquinta, 2000: Cirrus crystal terminal velocity. *J. Atmos. Sci.*, **57 ,** 916–938

Line 390: why does adding more realistic and complex ice habits alter the radiative heating of clouds? Please elaborate in the text to make it clear for the reader.

The following details have been added to the manuscript:
"Radiative properties of cirrus depend on the optical properties of its ice crystals, which in turn depends on the microphysical properties such as particle shape, particle size distribution, ice water content (eg., Key et al., 2002) and number concentration (eg, Liu et al., 2003). Previous studies (eg. *Takano and Liou* 1989) found that single-scattering albedo is related to aspect ratio and absorption coefficient of ice crystals. Also, Fu (1996) calculated the single-scattering properties of columnar and planar habits, using the geometric-optics-integration-equation technique [*Yang and Liou*, 1996b]. Key et al (2002) quantified significant sensitivity of ice cloud radiative properties to ice crystal optical properties of seven ice particle shapes.

The current study reveals a significant change in these microphysical properties across ice habit (Fig 9, Fig 11, and Fig 14). Additionally, ice crystal capacitance is parameterized differently from one ice habit to another depending on the maximum diameter of the crystals. This affects the growth rate of ice crystals differently, which in turn affects microphysical processes to alter the size distribution of ice crystals (Fig. 14), further modifying radiative heating."

Fu, Q.: An accurate parameterization of the solar radiative properties of cirrus clouds for climate models, J. Climate, 9, 2058– 2082, 1996.

Key, J. R., Yang, P., Baum, B. A., and Nasiri, S. L.: Parameterization of shortwave ice cloud optical properties for various particle habits, J. Geophys. Res., 107, doi:10.1029/2001JD000742, 2002.

Liu, H. L., Wang, P. K., and Schlesinger, R. E.: A numerical study of cirrus clouds. Part I: Model description, J. Atmos. Sci., 60, 1075–1084, 2003.

Takano, Y. and Liou, K. N.: Solar radiative transfer in cirrus clouds. I. Single-scattering and optical properties of hexagonal ice crystals, J. Atmos. Sci., 46, 3–19, 1989.

Yang, P. and Liou, K. N.: Geometric-optics – integral-equation method for light scattering by nonspherical ice crystals, Appl. Optics, 35, 6568–6584, 1996.

Line 419: "As the properties of cirrus clouds may differ depending on the geographical location and proximity to convections, this study is associated with a deep convection case during Asian summer monsoon." Please reword this statement as the first clause does not motivate the second. Also, add an article before "Asian summer monsoon".

Reworded as follows in line 466:
"The properties of cirrus clouds may differ depending on geographical location and proximity to convection. This study focuses on a specific case study associated with deep convection during the Asian summer monsoon."

Line 427: I don't think the sensitivity to grid spacing was really that clear from the figures.
Really, what is more important is the difference between the three schemes. This is in line with my comments of Figure 7, 8, and 9 below.

We agree that is not always significantly sensitive to change in grid spacing, except for Ishmael scheme, especially from 3k to 0.3km. We reworded the sentence to highlight is restriction as follows in lines 473-476

"Comparison of simulated microphysical properties to Stratoclim measurements shows that simulated HIWC in the TTL is significantly sensitive to horizontal grid spacing only with Ishmael scheme, with HIWC increasing substantially as grid spacing decreases from 3 to 0.3 km."

Line 433: change "vertical grid spacing" to "vertical resolution" if you want to keep this sentence. Otherwise, it reads as random and out of place relative to the rest of the paragraph. Consider placing it somewhere else.

Changed as suggested

**Figures**
General comment on all figures: those with subplots should include a label (a, b, c...) to help distinguish them within the main text. In some figures you refer to the Jensen scheme and other you label it as Ishmael. Please use a consistent label. Moreover, please chose a different colormap than Jet where applicable. This is not colorblind friendly, it introduces bias in interpretation of the results, and in some cases the reader cannot distinguish between the modelled data and the observations as they are plotted in the same color (e.g., Figure 10).

Corrected as suggested: All figures with subplot include labelings (a,b,c,...)

Corrected: Jensen → Ishmael

Corrected: All figures are tested for color blindness and colormap adjusted accordingly.

Old figure 13 -> new figure 11 , for the distribution of N_ice for each habit color of aggregates distribution and 99th and max values are changed to be color-blind friendly. And distinguish between habits.

Figure 1: this is a great figure. Though, please describe everything in the figure in the caption. For example, next to the arrows are the altitude of the aircraft and the distance, which I assume is the distance from the origin, but I can't tell for sure.

The following details are added to the caption of Figure 1:
"The corresponding altitude and distance from origin are indicted near each arrow (upper)"

Figure 4: this is a confusing figure as it looks very much like a tephigram. The caption should be much more descriptive, stating something along the lines of vertical resolution as a function of model vertical level. Also include a description of the shaded area. Also, do not refer to it as "vertical spatial resolution" as spatial indicates a single level and may confuse readers.

Corrected as suggested:
The new caption: "Figure 4: Vertical resolution as a function of model vertical level. The blue shaded area represents the TTL"

Figure7, 8, and 9: I am not these three figures and the associated discussion really portray the message the authors would like to convey, which is to show sensitivity to model domain. I am not sure this is really needed for this study. I would rather show the differences in the inner domain between the three microphysics schemes in one figure and move these three figures as they are now to an Appendix.

As the reviewer suggested, Figure7, 8, and 9 are shifted to Appendix and new Figure 7 is created with only profiles from the domain D04. (a) for Ishmael scheme, (b) for P3 scheme and (c) for Thompson scheme.

Figure 14: please move the legend below and make it larger so it is easier to read.

Modified as suggested. Also, additional description has been added to include the level of cold point temperature.

Figure 15: I would show the full range to be in line with the discussion on Line 266 and then perhaps place a box over the region of interest (0-50 um) and shown a blow up of that range as a subplot.

To be in line with the discussion, we added the 99$^{th}$ values on the figure 13 itself.

Figure 19: can the authors add fit lines to each subplot to aid their discussion in the paragraph starting on line 357?

As suggested: Fit lines with corresponding equations and correlation coefficients have been added to each subplot of figure 17 (old figure 19).

**Response to Minor comments:**

Sometimes you are missing an article is cases where you need one. For example, on Line 442 "and Ishmael scheme…" should read "and the Ishmael scheme…". Please review all instances of this.

Corrected throughout the manuscript as suggested.

Line 418 change "fewer" to "few"

Changed as suggested

Line 419 change "significantly" to "frequently"

Changed as suggested

Line 373: delete "the" before "controlling"

Changed as suggested

Line 458: make hydration lowercase.

Changed as suggested

---

## Author Response (AR2)

**Response to comments from Anonymous Referee #1**

We would like to thank the reviewer for the constructive suggestions, which helped us improve the clarity of the paper. Based on the comments and suggestions, we have revised the manuscript.
Please find below our point-by-point responses (in blue) to the reviewer.
Thank you again for your time in assessing our work.

According to the referees' reports, I am pleased to make a decision that your manuscript can be accepted for publication subject to technical corrections. Please read through your manuscript carefully, check for typing errors, and polish the English, as suggested by Referee #1. For instance,
Line 17: 'The lack' might be changed to 'However, the lack'
Line 18: 'is the investigation of' might be changed to 'of this study is to investigate'
Line 21: 'from' might be changed to 'over'
Line 33: 'and aggregates' might be changed to 'with aggregates'

**Response to Referee #1**

Line 17: 'The lack' might be changed to 'However, the lack'
Corrected
Line 18: 'is the investigation of' might be changed to 'of this study is to investigate'
Corrected
Line 21: 'from' might be changed to 'over'
Corrected
Line 33: 'and aggregates' might be changed to 'with aggregates'
Corrected